# Objective drives the consistency of representational similarity across datasets

Laure Ciernik [1 2 3]   Lorenz Linhardt [* 1 4]   Marco Morik [* 1 4]   Jonas Dippel [1 4 5]   Simon Kornblith [6]
Lukas Muttenthaler [1 4 5]

## Abstract

The Platonic Representation Hypothesis claims that recent foundation models are converging to a shared representation space as a function of their downstream task performance, irrespective of the objectives and data modalities used to train these models (Huh et al., 2024). Representational similarity is generally measured for individual datasets and is not necessarily consistent across datasets. Thus, one may wonder whether this convergence of model representations is confounded by the datasets commonly used in machine learning. Here, we propose a systematic way to measure how representational similarity between models varies with the set of stimuli used to construct the representations. We find that the objective function is a crucial factor in determining the consistency of representational similarities across datasets. Specifically, self-supervised vision models learn representations whose relative pairwise similarities generalize better from one dataset to another compared to those of image classification or image-text models. Moreover, the correspondence between representational similarities and the models' task behavior is dataset-dependent, being most strongly pronounced for single-domain datasets. Our work provides a framework for analyzing similarities of model representations across datasets and linking those similarities to differences in task behavior.

## 1. Introduction

Representation learning has seen remarkable progress in recent years, with state-of-the-art models achieving or even surpassing human-level performance in a wide range of computer vision tasks (Radford et al., 2021; Dehghani et al., 2023; Oquab et al., 2024; Zhai et al., 2023; Muttenthaler et al., 2024). With this progress in task performance, model representation spaces tend to approach each other irrespective of the models' training data or architecture (Huh et al., 2024). Even between models trained on different modalities, better task performances result in more similar representations. Two questions naturally arise from this phenomenon: *Do pairwise similarities of model representations transfer from one dataset to another, and are they generally linked to the models' downstream behavior?*

Investigating the correspondence between representation and behavior has a long history in representation learning and adjacent fields (e.g. Hermann & Lampinen, 2020; Sucholutsky et al., 2023). Two models can have different intermediate layer representations even though they show the same task behavior (Muttenthaler et al., 2023; Lampinen et al., 2024). However, if two models show different behavior, one can be certain that the output layer representations are different, since identical output representations map to the same behavior. What remains unclear is how behavior (e.g., downstream task accuracy) is affected by the convergence of model representations (i.e., the similarity of representations). Does this relationship depend on the similarity measure at hand, or may it be due to the nature of a dataset?

The field lacks consensus on *how to define (pairwise) representational similarity* (cf. Kriegeskorte et al., 2008; Kornblith et al., 2019a; Williams et al., 2021; Duong et al., 2023; Sucholutsky et al., 2023) and *how to systematically measure its relationship to behavior* (cf. Geirhos et al., 2021; Muttenthaler et al., 2023; Sucholutsky et al., 2023; Lampinen et al., 2024). Here, we use similarity measures that are widely used and have been empirically proven to be useful for measuring representational similarity of neural networks, such as Centered Kernel Alignment (CKA; Kornblith et al., 2019a; Raghu et al., 2021), and Representational Similarity Analysis (RSA; Kriegeskorte et al., 2008).

---

[*]Equal contribution   [1]Machine Learning Group, Technische Universität Berlin, Berlin, Germany [2]Hector Fellow Academy, Karlsruhe, Germany [3]European Laboratory for Learning and Intelligent Systems (ELLIS), Tübingen, Germany [4]BIFOLD—Berlin Institute for the Foundations of Learning and Data, Berlin, Germany [5]Aignostics, Berlin, Germany [6]Anthropic, California, United States of America. Correspondence to: Laure Ciernik <ciernik@tu-berlin.de>, Lukas Muttenthaler <lukas.muttenthaler@tu-berlin.de>.

*Proceedings of the 42nd International Conference on Machine Learning*, Vancouver, Canada. PMLR 267, 2025. Copyright 2025 by the author(s).

However, pairwise representational similarity is usually measured for a single dataset and not across datasets. It may be that the representations of two vision models are highly similar for a dataset of raccoons but dissimilar for a dataset of sunflowers — due to their different *domains*. Similarly, the representations of two models may exhibit high similarity for satellite data but show low similarity for images of fruit — because of the images' different *structure*.

Thus, it appears crucial to scrutinize the factors that determine pairwise representational similarity across sets of stimuli. Knowing about those factors will enable us to generalize pairwise representational similarity across datasets and more effectively find model sets that have learned a similar representation of the world. For a large set of diverse vision models, we examine their pairwise representational similarities for both structured and unstructured datasets from different domains. Our contributions and findings are as follows:

- We propose a principled way of measuring the consistency of pairwise similarity across datasets, by measuring whether the (relative) pairwise similarities of model representations from one dataset are transferable to the (relative) pairwise similarities of another dataset.

- We find that the objective function is a crucial factor for determining the consistency of pairwise representational similarities across datasets, whereas architecture and model size do not appear to be major factors. In particular, SSL models show more reliable generalization across stimulus sets compared to image-text and supervised models.

- Our findings show the same trend across different similarity measures, including those that emphasize local versus global representational structure.

- Pairwise similarities of model representations strongly correlate with the models' differences in task performance for single-domain datasets, while multi-domain datasets show highly variable and specialized datasets consistently low correlations.

## 2. Related work

**Measuring the similarity of deep neural networks**. Representational similarity measures quantify the degree of overlap between the representations of two models when processing the same input. They have been used as a tool to understand deep neural networks (Li et al., 2015; Nguyen et al., 2021; Mehrer et al., 2020), investigate human-machine alignment (Sucholutsky et al., 2023; Xu & Vaziri-Pashkam, 2021), and as an objective for model distillation (Tian et al., 2020; Saha et al., 2022; Zong et al., 2023). Different *representational similarity* measures have been proposed, includ-

ing Canonical Correlation Analysis (CCA; Hotelling, 1992; Morcos et al., 2018), Singular Vector Canonical Correlation Analysis (SVCCA; Raghu et al., 2017), the previously mentioned CKA and RSA, among others (e.g., Williams et al., 2021; Ding et al., 2021; Barannikov et al., 2022; Cui et al., 2022). They can be distinguished from *functional similarity* measures, such as performance difference (Klabunde et al., 2025), error consistency (Geirhos et al., 2020b), and true-positive agreement (Hacohen et al., 2020), which are based on model outputs rather than internal representations (Klabunde et al., 2025), or model stitching (Bansal et al., 2021; Csiszárik et al., 2021; Merullo et al., 2023), which measures compatibility, not similarity, of internal representations (Hernandez et al., 2022).

Importantly, similarity measures may differ in the invariances they evoke and the properties of representational spaces they capture (Klabunde et al., 2025). For example, local structures may be better captured by measures based on nearest neighbors (Huh et al., 2024) or localized kernels (Kornblith et al., 2019a), while capturing global similarities may be better achieved by wide kernels. Here, we make use of kernel-based CKA, as it allows probing both local and global representation structures by varying the kernel, does not require the compared representations to be of the same dimension, and is widely used (e.g., Raghu et al., 2021; Maniparambil et al., 2024; Saha et al., 2022; Zong et al., 2023; Ding et al., 2019; Baratin et al., 2021).

**Drivers of similarity in vision models**. In contrast to language processing systems (e.g., Achiam et al., 2023; Team et al., 2023), vision models are trained with a wide range of different learning objectives. These include (a) supervised learning objectives using classification datasets such as ImageNet (Krizhevsky et al., 2012b); (b) SSL objectives with a reconstruction or multi-view matching loss (Chen et al., 2020a; Caron et al., 2021; He et al., 2022); and (c) objectives for jointly learning image and text representations (Radford et al., 2021; Zhai et al., 2023). This makes them interesting candidate models for analyzing representational similarities. The trend to larger model sizes (e.g. Dehghani et al., 2023) combined with increased dataset sizes (e.g. Schuhmann et al., 2022; Oquab et al., 2024) and multi-modality appears to be leading to a convergence of model representations (Huh et al., 2024). Similarity between otherwise identical networks trained from different random initializations increases with width (Morcos et al., 2018; Kornblith et al., 2019a), as predicted by theoretical studies of properties of neural networks in the infinite width limit (Lee et al., 2018; Matthews et al., 2018; Yang & Hu, 2021). Huh et al. (2024) hypothesize that convergence is further driven by task generality and inductive bias. Other works find that architecture and dataset (Raghu et al., 2021) play a central role in determining the representational structure of a model.

**Consistency of similarity**. Beyond the similarity function, representational comparisons depend on both the extraction layer (Kornblith et al., 2019a; Raghu et al., 2021) and the (probe) dataset (Cui et al., 2022). For example, measuring similarity for a subset of a data distribution can yield different similarities than measuring it for the whole dataset, albeit maintaining the ranking of similarities between models (Brown et al., 2024). However, recent work demonstrated that representational similarities (e.g., measured via RSA) can be used to effectively select (downstream) task-specific models (Dwivedi & Roig, 2019; Borup et al., 2023). From a behavioral perspective, it has been shown that the ranking of the *agreement* of ImageNet classifier pairs generalizes to out-of-distribution data (Baek et al., 2022). Due to the limited scope of prior work (e.g., only studying a few models (Klabunde et al., 2023; Brown et al., 2024), the driving factors of the *consistency* of representational similarities across vision datasets have not yet been identified.

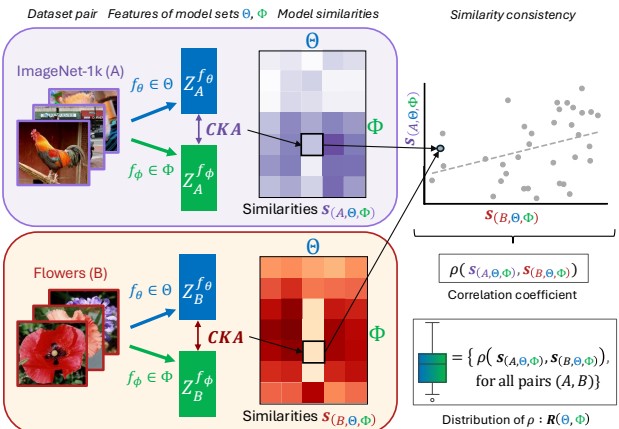

Figure 1: Pairwise similarity analysis framework. Let $\boldsymbol{A}$ and $\boldsymbol{B}$ be two sets of stimuli and $\Theta$ and $\Phi$ be two sets of models. For each dataset, we use all models within the two sets to extract representations, e.g., $\boldsymbol{Z}_A^f$ being the features extracted from dataset $\boldsymbol{A}$ using model $f$. Subsequently, for each pair of models $(f_\theta, f_\phi)$, where $f_\theta \in \Theta$ and $f_\phi \in \Phi$, we compute the CKA similarity between their representations (i.e., $CKA(\boldsymbol{Z}_A^{f_\theta}, \boldsymbol{Z}_A^{f_\phi})$), yielding a similarity vector $\boldsymbol{s}_{(A,\Theta,\Phi)}$. This vector can be displayed as a matrix, where each entry represents the similarity between two models for a single dataset. In the scatter plot, we contrast two such vectors computed on the same model sets but evaluated on different datasets $\boldsymbol{A}$ and $\boldsymbol{B}$. The Pearson correlation coefficient $\rho$ between the similarities quantifies the consistency of similarities across the two datasets. The distribution of $\rho$ across all dataset pairs, $\boldsymbol{R}(\Theta, \Phi)$, indicates the consistency of (relative) representational similarities across stimuli.

# 3. Methods

In this section, we propose a novel approach for analyzing the consistency of representational similarities across multiple datasets. Fig. 1 illustrates this framework.

**Extracting representations**. We are interested in comparing the similarities of different vision models. We perform this comparison by extracting their latent representations and examining them. Let $f_\theta : \mathbb{R}^d \mapsto \mathbb{R}^p$ be a pretrained neural network function parameterized by a fixed set of parameters $\theta$ that maps the $d$-dimensional images to $p$-dimensional vector representations. Let $\boldsymbol{X} \in \mathbb{R}^{n \times d}$ be a dataset of $n$ stacked images. For each image $\boldsymbol{x} \in \mathbb{R}^d$ in a dataset, we extract its corresponding latent representations $f_\theta(\boldsymbol{x}) = \boldsymbol{z} \in \mathbb{R}^p$. For supervised models, we typically use the penultimate layer to extract this representation; for self-supervised vision models, the average pooling layer; and for image-text models, the image encoder. We denote by $\boldsymbol{Z} \in \mathbb{R}^{n \times p}$ the matrix of stacked vector representations, which is unique for each model and dataset combination.

**Computing representational similarities for a dataset**. We compute the similarity between two models $f_\theta$ and $f_\phi$ for a dataset by applying CKA to their vector representations $\boldsymbol{Z}^{f_\theta}$ and $\boldsymbol{Z}^{f_\phi}$. CKA computes the similarity based on the normalized Hilbert-Schmidt Independence Criterion (Gretton et al., 2005), applied to the kernel matrices of both representations. Using CKA with a linear kernel focuses on global similarity structure, while an RBF kernel with small $\sigma$ measures local similarity structure (Kornblith et al., 2019a; Alvarez, 2023). We compared their behavior (shown in Fig. 3 and in Appx. F) and observed similar trends for both kernels. Therefore, we use CKA with a linear kernel for the remainder of this paper if not mentioned otherwise.

Computing the similarity requires the full kernel matrix over the dataset, which scales quadratically with the number of data points. Hence, the exact computation becomes intractable for large datasets. However, we demonstrate in Appx. C that for CKA linear a diverse subset of 10,000 samples is sufficient to accurately estimate the similarities of two models for an ImageNet-scale dataset.

**Representational similarities across datasets**. Because image representations depend on both a model and a dataset, the representational similarities between two models may vary across different datasets. To quantify how consistent these similarities remain across datasets (i.e., the transferability of similarity relationships), we compute the correlation coefficient for pairwise model similarities (measured for a large set of model pairs) between datasets. We can then analyze these correlations for specific subsets of models grouped by their training factors.

Specifically, we scrutinize how the pairwise similarities

of models from two model sets, $\Theta$ and $\Phi$, correspond between two datasets, $\boldsymbol{A}$ and $\boldsymbol{B}$. By computing the pairwise representational similarities between all model pairs $(f_\theta, f_\phi)$, separately for each dataset, where $f_\theta \in \Theta$, $f_\phi \in \Phi$ and $f_\theta \neq f_\phi$, we obtain a similarity vector $\boldsymbol{s} \in \mathbb{R}^k$, where $k$ denotes the number of distinct model combinations. A similarity vector for two sets of models and a dataset $\boldsymbol{X}$ is then defined as $\boldsymbol{s}_{(\boldsymbol{X}, \Theta, \Phi)} = \left( \text{CKA}(\boldsymbol{Z}_{\boldsymbol{X}}^{f_\theta}, \boldsymbol{Z}_{\boldsymbol{X}}^{f_\phi}) \mid f_\theta \in \Theta, f_\phi \in \Phi, f_\theta \neq f_\phi \right)^\top \in \mathbb{R}^k$.

To quantify the consistency of similarities between two datasets $\boldsymbol{A}$ and $\boldsymbol{B}$, we use the Pearson correlation coefficient between the similarity vectors $\boldsymbol{s}_{(\boldsymbol{A}, \Theta, \Phi)}$ and $\boldsymbol{s}_{(\boldsymbol{B}, \Theta, \Phi)}$, i.e., $\rho\left( \boldsymbol{s}_{(\boldsymbol{A}, \Theta, \Phi)}, \boldsymbol{s}_{(\boldsymbol{B}, \Theta, \Phi)} \right)$. The Pearson correlation measures the degree to which the similarity trends between models are preserved across datasets, focusing on the relative positioning of model pairs rather than the absolute similarity values. In other words, if two models $f_\phi$ and $f_\theta$ are more similar than $f_\phi$ and $f_\psi$ on one dataset, a high Pearson correlation coefficient indicates that this relationship transfers to the other dataset. Conversely, a low correlation value suggests that the models' relationships shift significantly depending on the dataset, showing variability in processing different data distributions. Thus, the Pearson correlation coefficient is a meaningful indicator of how dataset-dependent or dataset-invariant the model similarities are.

To assess the consistency of these correlations across dataset pairs, we examine the distribution of correlation coefficients for all possible dataset pairs within the set of available datasets $\mathcal{D}$ : $\boldsymbol{R}(\Theta, \Phi) = \left\{ \rho\left( \boldsymbol{s}_{(\boldsymbol{A}, \Theta, \Phi)}, \boldsymbol{s}_{(\boldsymbol{B}, \Theta, \Phi)} \right) \mid \boldsymbol{A}, \boldsymbol{B} \in \mathcal{D}, \boldsymbol{A} \neq \boldsymbol{B} \right\}$. The spread of the correlation coefficients across all dataset pairs indicates the consistency of the relative pairwise model similarities (*similarity consistency*).

# 4. Experiments

Our experiments investigate the consistency of representational similarities between vision models across various datasets using our framework. Following the experimental setup (§4.1), we first analyze how model similarities transfer across datasets (§4.2, 4.4) and find consistent model groups through similarity clustering (§4.3). We then investigate how model and dataset characteristics influence similarity consistency (§4.5 and 4.6), and conclude by examining the relationship between representational similarities and models' performance differences on downstream tasks (§4.7). The code and the data to run our analyses and reproduce the experimental results are publicly available at https://github.com/lciernik/similarity_consistency.

## 4.1. Setup

**Models**. We evaluated 64 vision models, representing a diverse range of architectures, training paradigms, and parameter scales. We focused on general-purpose vision models trained on large-scale datasets (minimum ImageNet-1k size) with broad semantic diversity, excluding models trained on specialized or synthetic datasets. A complete list of models and their characteristics can be found in Appx. A.

**Model grouping**. Instead of focusing on the representational similarities of individual model pairs, we analyze aggregated similarities between sets of models. We categorize models based on four attributes: *training objective*, *training data (size)*, *model architecture*, and *model size* (see Tab. 3). This allows us to examine how different model characteristics relate to the behavior of representational similarities while acknowledging that these attributes are often correlated across models (see Fig. 9).

**Datasets**. We evaluated pairwise model similarities across 20 datasets from the CLIP benchmark (Cherti & Beaumont, 2022) and 3 datasets from Breeds (Santurkar et al., 2021). This set includes various VTAB datasets as well as ImageNet-1k. Following the categorization proposed in (Zhai et al., 2020), we classified the datasets into three main types: natural (e.g., ImageNet-1k), specialized (e.g., PCAM), and structured (e.g., DTD) image datasets. Furthermore, we partition the natural image datasets into single- and multi-domain categories. A list of all datasets can be found in Tab. 1.

## 4.2. Do representational similarities transfer across datasets?

The transferability of model similarities across diverse datasets is crucial for model comparison and selection. If similarities are consistent across datasets, we can evaluate models on a single dataset and generalize their relationships to other datasets. We analyzed this by computing pairwise CKA similarities for all model pairs across our datasets.

Model similarities (e.g., the range of values and the similarity trends) appear to vary across datasets (Fig. 2, Appx. D). We observe the highest similarity within image-text models (yellow boxes), while a group of self-supervised models (white boxes) shows consistent similarity trends across datasets. Yet, there are also model groups with higher variability (cyan boxes), mainly containing supervised models. The standard deviation matrix (rightmost matrix in Fig. 2) quantifies these differences, indicating that model similarities do not transfer uniformly across datasets.

To further investigate the transferability of model similarities across datasets, we examine the relationship between the mean and standard deviation of similarity values. The left panel in Fig. 3 shows an inverse U-shape trend (fur-

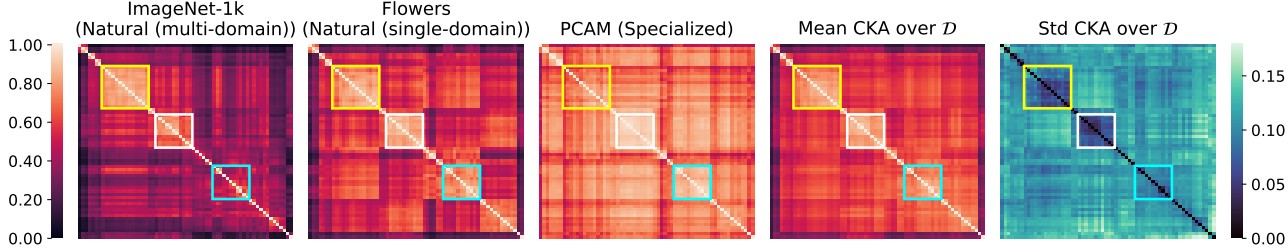

Figure 2: Representational similarity using linear CKA. Left to right: natural multi- and single-domain, and specialized datasets, followed by mean and standard deviation across all datasets. Models (rows and columns) are ordered by a *hierarchical clustering* of the mean matrix. Yellow and white boxes highlight regions with more stable similarity patterns across datasets, corresponding to some image-text (yellow) and self-supervised model pairs (white), while cyan boxes show higher variability for mainly supervised model pairs.

ther analyzed in Appx. E): model pairs with extreme mean similarities exhibit low variability, while variability for mid-range similarities is higher. To verify that this observed variability is not an artifact of the similarity metric, we compare the mean similarity values obtained using linear CKA with two other metrics: CKA with an RBF kernel where $\sigma = 0.2$ (a local similarity measure, cf., Appx. F for $\sigma$ selection) and RSA using Spearman correlation (a global similarity measure). The high correlations among these measures confirm that mid-range similarity values consistently vary across metrics, indicating limited stability across datasets.

Given this variability, we investigate persistent trends in model relationships across datasets by addressing two questions: a) Can we identify subgroups of models whose similarity to other models (consistently) cluster across different datasets? b) Do the *relative* representational similarities within or between these subgroups remain stable/consistent across datasets? Answering these questions will allow us to make generalizable, albeit potentially weaker, claims about models' relationships from a single data source.

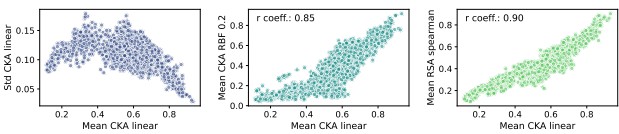

Figure 3: Variability of representational similarities across datasets. **Left:** Mean versus standard deviation of pairwise similarities. **Center/Right:** Mean similarity values using CKA RBF ($\sigma = 0.2$) or RSA as a function of linear CKA. High Pearson correlation coefficients indicate consistency across different similarity metrics.

### 4.3. Do representational similarities cluster according to model categories?

We qualitatively assess the alignment between our predefined model categories (§4.1) and the clustering patterns

observed in the t-SNE embedding of the model-model similarity matrices, as illustrated in Fig. 4. The training objective appears to be the most distinctive categorization, with all image-text and most SSL models forming distinct clusters across all three datasets. The objective and dataset categories show substantial overlap, potentially confounding further analysis. For instance, only image-text models have been trained on XLarge datasets, while all models trained on ImageNet-21k are supervised models (Appx. A). Additionally, no image-text model has been trained on ImageNet-1k or ImageNet-21k. In contrast, no obvious clustering is discernible based on architecture or model size. These observations provide initial evidence that the training objective might play a key role in representational similarity.

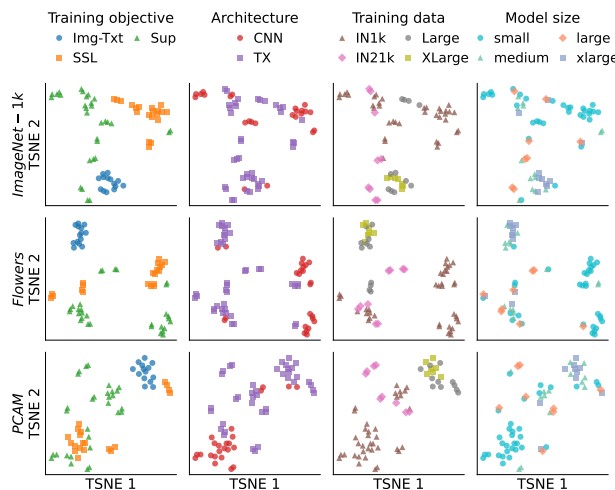

Figure 4: t-SNE visualization of model similarity matrices for ImageNet-1k, Flowers, and PatchCamelyon (PCAM) datasets. Embeddings are color-coded by model attributes: training objective, architecture, training data, and model size.

### 4.4. Do relative representational similarities remain consistent across different datasets?

Given the observation that representational similarities are not directly transferable across datasets but depend on both the models and the dataset, we now examine the consistency of *relative* representational similarities across dataset pairs. Following the method described in §3, the top row of Fig. 5 reveals positive correlations between pairwise model similarities across three exemplary dataset pairs. Across all dataset pairs, the mean Pearson correlation coefficient is 0.756 (std=0.124). This indicates that the ordering of pairwise model similarities is largely consistent across datasets, suggesting a degree of transferability in the relationships of representational similarities. However, the high standard deviation and the distributions of similarities shown in Fig. 5 suggest the existence of model-pair groups that may exhibit distinct consistencies of similarity values.

The second row in Fig. 5 shows model pairs within the same training objective, highlighting differences in group-specific consistency levels. We find that SSL models show the strongest similarity consistency across datasets, image-text models form a cluster of generally high similarities, and supervised models achieve the weakest consistency. In the following section, we will analyze the aggregated measure across all dataset pairs and revisit all categories in detail.

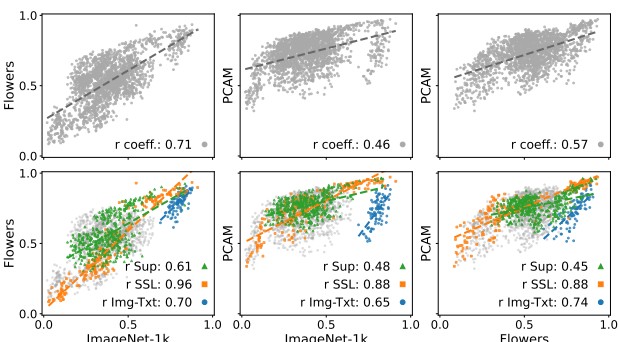

Figure 5: Pairwise representational similarities for model pairs across three dataset combinations: ImageNet-1k vs. Flowers, ImageNet-1k vs. PCAM, and Flowers vs. PCAM. **Top row:** Pearson correlation coefficients for all model pairs. **Bottom row:** Same data, with colored points highlighting model pairs within the same training objective category ($\Phi_{\text{Img-Txt}}$, $\Phi_{\text{SSL}}$, and $\Phi_{\text{Sup}}$).

### 4.5. Which model categories influence similarity consistency?

Fig. 6 shows the distribution of Pearson correlation coefficients $R(\Theta, \Phi)$ for each combination of model sets $(\Theta, \Phi)$ across all dataset pairs.

**Training objective**. Self-supervised models ($\Phi_{\text{SSL}}$) show

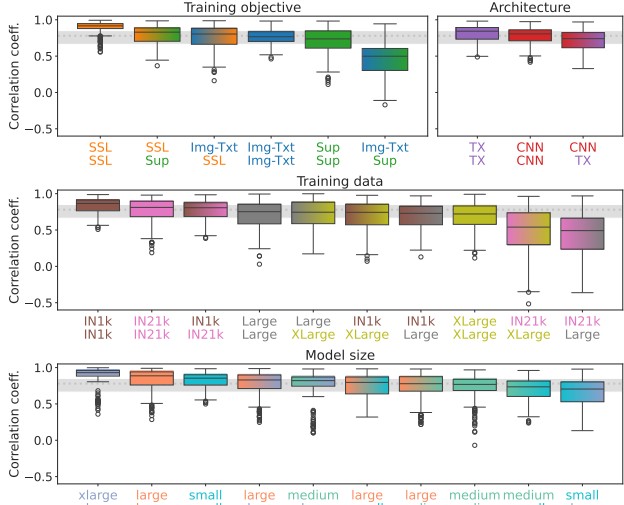

Figure 6: Distribution of correlation coefficients for model similarities across dataset pairs (consistency values). Each subplot represents a model category: training objective, architecture, training data, and model size. Within each category, distributions show the similarity consistencies between model set combinations (e.g., $R(\Phi_{\text{Img-Txt}}, \Phi_{\text{Sup}})$). The boxes in each subplot are sorted in decreasing median correlation. The dotted line indicates the overall median, while the gray area spans the 25th to 75th percentiles of correlations across all model pairs (no category consideration).

the highest similarity consistency, even when compared to models with different training objectives. This suggests that their relative similarities are most transferable across datasets, likely due to the ability of SSL to capture dataset-independent features. In contrast, supervised models show the weakest correlations, particularly when compared to image-text models ($\Phi_{\text{Img-Txt}}$, $\Phi_{\text{Sup}}$), indicating a less reliable transfer of relative pairwise similarities across datasets.

**Model architecture**. While no significant differences in similarity consistency exist between transformers and convolutional networks, comparisons across architectures show slightly lower consistency, potentially indicating distinct architectural inductive biases.

**Training data**. We find a low median correlation and a high variance for the model set pairs ($\Phi_{\text{IN21k}}, \Phi_{\text{Large}}$) and ($\Phi_{\text{IN21k}}, \Phi_{\text{XLarge}}$). However, these effects may be confounded by the training objectives, as both $\Phi_{\text{Large}}$ and $\Phi_{\text{XLarge}}$ are mainly image-text models, while $\Phi_{\text{IN21k}}$ consists of supervised models. On the other hand, $\Phi_{\text{IN1k}}$, which includes models trained on an ImageNet-21k-like dataset but with diverse training objectives, does not show the same trend. Consequently, training data (size) appears to have a less significant impact on the consistency of similarities than other factors. However, this applies to models trained on large and diverse datasets; as shown in Appx. I, training

on less diverse data noticeably affects consistency.

**Model size**. Models in the same size category show a higher similarity consistency than models of different size categories (e.g., $\Phi_{\text{small}}$ and $\Phi_{\text{xlarge}}$). This indicates model size influences the transferability of relative similarities across datasets, though less significantly than training objective.

Our analysis identifies the training objective as a primary factor influencing similarity consistency, with model size playing a less pronounced role. Self-supervised models consistently demonstrate above-average correlations across datasets. The effects of model architecture and training data (when mainly considering dataset size) are minimal, suggesting that these factors are less critical in determining the transferability of relative similarities across datasets (see Appx. G for an analysis using single-element model sets controlling for model architecture and size).

### 4.6. Do dataset categories influence similarity consistency?

Previously, we analyzed similarity consistency from the perspective of models, noting substantial variation in consistency across all dataset pairs. In this section, we shift our focus to identifying groups of datasets that exhibit more stable relative similarities, aiming to uncover variables that might explain these variations.

The top panel in Fig. 7 shows the Pearson correlations of all model pairs for each dataset pair individually, i.e., $\rho\left(s_{(A,\Phi_{\text{All}},\Phi_{\text{All}})}, s_{(B,\Phi_{\text{All}},\Phi_{\text{All}})}\right)$ for all datasets pairs $(A, B)$. The consistency of representational similarities varies strongly across dataset pairs, showing higher consistency within dataset categories than across them, as exemplified by high consistencies within natural multi-domain datasets but lower ones when paired with specialized datasets. We hypothesize that high-consistency dataset categories largely share visual features. For example, the Breeds datasets, Caltech-101, Country-211, STL-10, and Pets contain (often-times centered) natural objects and scenes. For CIFAR-10 and CIFAR-100, the particularly high consistency may also be supported by their relatively low resolution. Moreover, some datasets are inconsistent with almost any other dataset, particularly FGVC Aircraft and medical imaging datasets (Diabetic Retinopathy and PCAM) – most likely due to their unique visual structure, which differs across medical domains. Interestingly, ImageNet-1k exhibits a milder yet significant pattern of inconsistency, in particular, in combination with CIFAR-10 and CIFAR-100.

To investigate whether specific dataset categories are responsible for the difference in consistency observed in the last section, we focus on the correlations for the model sets with the highest ($\Phi_{\text{SSL}}$, $\Phi_{\text{SSL}}$) and lowest ($\Phi_{\text{Img-Txt}}$, $\Phi_{\text{Sup}}$) consistency (see the bottom left and right panels of Fig. 7). The

two matrices reveal distinctly different patterns, highlighting how model categories influence similarity consistency.

For SSL model pairs, we generally observe high consistency across most datasets, with FGVC Aircraft being an exception, showing lower correlations with the rest. This confirms that SSL models tend to produce more transferable similarity structures across datasets. In contrast, image-text/supervised model pairs show relatively higher consistency for multi-domain than single-domain datasets, suggesting better similarity transfer when semantic categories in the dataset match the training data of the models. However, Pascal VOC 2007 appears as an outlier when compared to ImageNet-1k and its subsets, likely because its bounding-box-cropped images lack contextual information, causing single-domain-like behavior.

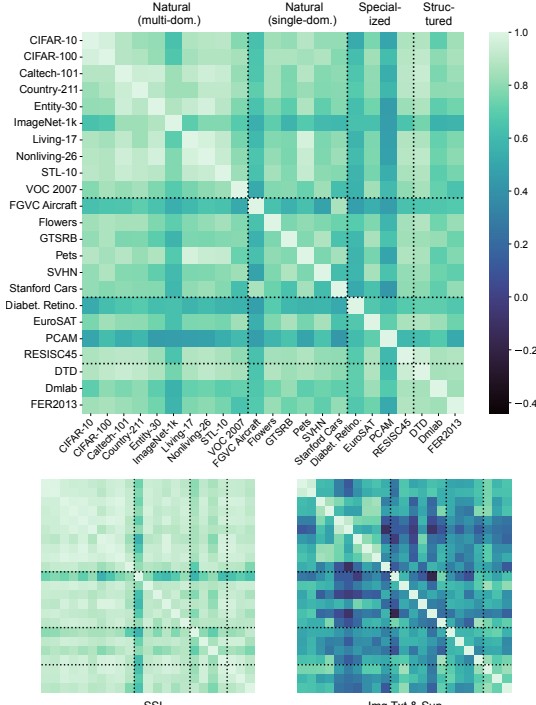

Figure 7: Pearson correlation of representational similarities for all dataset pairs computed on different model sets. **Top:** all available model pairs; **Bottom left:** SSL models ($\Phi_{\text{SSL}}$, $\Phi_{\text{SSL}}$); **Bottom right:** image-text and supervised models ($\Phi_{\text{Img-Txt}}$, $\Phi_{\text{Sup}}$). Dataset categories are delineated by dotted lines.

Overall, we observe the highest consistency within multi-domain image datasets, suggesting that models are consistent when dealing with diverse semantic categories. In contrast, we find weaker consistency for ($\Phi_{\text{Img-Txt}}$, $\Phi_{\text{Sup}}$) between multi-domain and single-domain datasets, indicating that even though these models effectively generalize across diverse datasets, their representational structures differ substantially when evaluated on single domains. Self-

supervised models do not exhibit this sensitivity to datasets; their similarity relationships appear to be more stimuli-invariant, maintaining consistent similarities regardless of whether the data is multi-domain or single-domain.

### 4.7. Can representational similarity predict performance gaps?

To investigate the relationship between downstream task performance and representational similarity, we contrast two measures for each dataset: (1) the CKA value for each model pair and (2) the absolute difference in classification performance, referred to as *performance gap* (see Appx. B for details on performance computation). Fig. 8 shows this relationship for one dataset per category (see Fig. 21 & 22 for additional datasets). While prior work suggests well-performing models produce similar representations (Huh et al., 2024), we find this relationship is dataset-dependent.

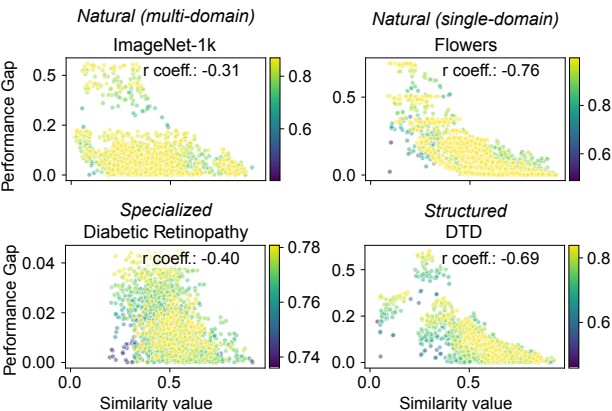

Figure 8: Model similarity (CKA linear) versus absolute difference in downstream task performance (top-1 accuracy) for model pairs across four dataset categories: natural multi-domain, natural single-domain, specialized, and structured. The color of each point indicates the downstream task accuracy of the better-performing model.

**Natural multi-domain** datasets, such as ImageNet-1k, show weak negative correlation between similarity and performance gap. Many model pairs achieve high performance despite low representational similarity (yellow dots in the lower left corner). This suggests that multi-domain datasets might provide richer contextual information, enabling multiple high-performance strategies that do not converge to similar representations. However, correlation coefficients in this dataset category show notable variability (see Appx. J).

**Natural single-domain and Structured** datasets, in contrast, exhibit a strong negative correlation between similarity and performance gap. The absence of high-performing, dissimilar model pairs suggests a close link between performance and representational similarity, indicating that

successful models in these datasets rely on capturing specific discriminative features for closely related classes. Observing this behavior in both structured and single-domain datasets suggests that models may depend on representing more structural attributes to distinguish between classes. However, the limited number of structured datasets restricts broader conclusions.

In summary, our analysis shows different trends depending on the dataset. Multi-domain datasets support a wider range of successful strategies, while performing well on single-domain and structured datasets appears to require a more limited set of features, resulting in stronger correlations between performance and similarity. This phenomenon aligns with findings from previous studies, which have shown that models can leverage contextual cues in multi-domain datasets to achieve strong performance without developing robust, generalizing features (Lapuschkin et al., 2019; Geirhos et al., 2019; 2020a).

## 5. Discussion

The representations of foundation models appear to converge to a canonical representation, irrespective of their training data and objectives (Huh et al., 2024). However, this may just be an artifact of the datasets that the community commonly uses for evaluation. Thus, in this paper, we presented a *systematic way of analyzing pairwise similarities of model representations across sets of stimuli*. This approach allowed us to examine whether representational similarities generalize across datasets and identify variables influencing their consistency.

We found that the *training objective* of models is a central factor for determining the consistency of pairwise representational similarities across datasets, whereas architecture and model size appear less important. Among training paradigms, image-text training leads to the most similar representation spaces, irrespective of the model's training data or architecture. For individual datasets, SSL model representations are not as similar among themselves as the representations of image-text models, but the pairwise similarities between SSL models and other models are highly consistent across different datasets.

We hypothesize that both image-text and supervised models learn distinct semantic categories during training that lead to representations whose features overfit to these categories and, thus, do not generalize well to datasets that do not contain them. In contrast, self-supervised pure vision models, which are not constrained to learn explicit semantic categories during training, may develop representations that better accommodate diverse stimuli. This leads to relative pairwise similarities that vary less across datasets than the similarities of image-text or supervised model representa-

tions.

Alternatively, the SSL model representations, usually extracted from the average pooling layer (cf., Chen et al., 2020b; Muttenthaler et al., 2023; Tian et al., 2024), may be further away from the models' task behavior than the image encoder and penultimate layer representations from image-text and supervised models. One benefit of our framework is that it can be applied to the representations of any model layer, allowing future work to test this hypothesis. Furthermore, in contrast to image-text and supervised models, the tasks used to train SSL models are extremely heterogeneous. This may yet be another reason for the higher consistency of representational similarities among SSL models.

Finally, we have shown that the pairwise similarities between model representations are only in part predictive of their differences in downstream task behavior. While we observe a strong correlation on datasets with limited contextual information, for others, e.g., ImageNet-1k, there is no obvious correspondence between the pairwise similarities of model representations and the models' differences in task performance. Moreover, the relationship between pairwise similarities and task performance is substantially different between natural multi-domain datasets. This is surprising in light of recent findings that have shown a convergence of model representations as a function of task performance (Huh et al., 2024) and a strong linear relationship between a model's ImageNet performance (i.e., performance on the training set distribution) and its downstream accuracy (Kornblith et al., 2019b). This suggests that the transferability of pairwise similarities across datasets follows a more complex relationship than the transferability of task performance. While different objectives and hyperparameters lead to similar task performances (Kornblith et al., 2021), they seem to change the similarity spaces of the learned representations in a way that affects the relationship between pairwise model similarities and differences in downstream behavior between datasets. Therefore, whether the claims of The Platonic Representation Hypothesis hold or not depends on the nature of a dataset rather than being true for all sets of stimuli.

**Conclusion.** In summary, we presented a blueprint for systematically analyzing pairwise representational similarities across different sets of stimuli. This allowed us to demonstrate that *pairwise similarities rarely transfer from one dataset* to another and depend on both the models' objective functions and the datasets' structure and domain. The community has long optimized for finding the optimal hyperparameters to improve task performance (e.g. Tolstikhin et al., 2021; Steiner et al., 2022; Liu et al., 2022; Dehghani et al., 2023) but neglected its ramifications for the similarity structure of the models' representation spaces. Only few recent works investigated how representations are affected

by training tasks (e.g. Lampinen et al., 2024; Kornblith et al., 2021). We believe that pinpointing the variables of model training affecting (dis-)similarities between model representations beyond the training dataset will ultimately help to build more interpretable systems with which humans are more likely to interact (Lake & Baroni, 2023; Sucholutsky et al., 2023; Muttenthaler et al., 2024; Tessler et al., 2024).

## Acknowledgments

LC is funded by the Hector Fellow Academy and is grateful for their support.

## Impact Statement

This work investigates the consistency of representational similarities across datasets and provides new insights into how stimuli and model characteristics interact. Our findings challenge assumptions about universal representational convergence and highlight the nuanced relationship between representations and downstream task performance, paving the way for more robust and aligned models. While these findings contribute to the advancement of machine learning, we do not foresee any immediate negative societal consequences requiring specific emphasis.

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

# Appendix

## A. Models and Datasets

We considered 23 downstream datasets (Tab. 1) and 64 pretrained vision models (Tab. 2) in our analysis. For each dataset, we selected the training split, potentially subsetted for large datasets as described in Appx. C, to compute the model representational similarities, while we use the validation or test split to compute downstream performance.

Table 1: Datasets of clip-benchmark and breeds and their domains according to (Zhai et al., 2020) with additional separation of natural images into multi- and single-domain.

| Natural (multi-domain) | Natural (single-domain) | Specialized | Structured |
|---|---|---|---|
| Caltech101 | Cars | Diabetic Retinopathy | FER2013 |
| CIFAR-10 | FGVC Aircraft | EuroSAT | Dmlab |
| CIFAR-100 | Flowers | PatchCamelyon (PCam) | DTD |
| Country211 | GTSRB | RESISC45 | |
| ImageNet-1k | Pets | | |
| STL-10 | SVHN | | |
| Pascal VOC 2007 | | | |
| Entity-30 (Breeds) | | | |
| Living-17 (Breeds) | | | |
| NonLiving-26 (Breeds) | | | |

**Models**. The models were chosen to represent a diverse set of training datasets, training objectives, model sizes, and architectures. Training objectives include standard supervised learning (Sup), self-supervised learning (SSL), and image-text alignment (Img-Txt). The models have been trained on datasets spanning four classes of increasing scale and semantic diversity. The most focused class consists of ImageNet-1k (IN1k; (Deng et al., 2009)) with 1,000 object categories, followed by models trained on ImageNet-21k (IN21k; (Ridnik et al., 2021)) or a combination of IN21k and IN1k, expanding to 21,000 categories. The "Large" class encompasses COYO-700M (Byeon et al., 2022), LAION400M (Schuhmann et al., 2021), LVD-142M (Oquab et al., 2024), and WIT-400M (Radford et al., 2021), which move beyond curated object categories to include broader visual concepts and text descriptions. Finally, the "XLarge" class includes LAION2B (English subset of LAION5B (Schuhmann et al., 2022)), Merged2B (merged version of LAION2B and COYO-700M; (Sun et al., 2023)), and WebLI (Chen et al., 2023), which contain billions of image-text pairs covering an extremely wide range of visual and semantic content. In terms of architecture, the models employ a range of designs, including Convolutional Neural Networks (CNNs) such as ResNet, EfficientNet, ConvNeXt, and VGG, as well as Transformer-based models like ViT and Swin-Transformer. Further, we divide the models into four different size categories, ranging from small, with around 12 million parameters, to xlarge, with over 1.4 billion parameters. Tab. 3 summarizes the number of models in each category, and Fig. 9 shows the distribution of models across pairs of categories, highlighting the relationships between different model attributes.

Table 3: Number of models in each attribute category.

| Category | Nr. models | Category | Nr. models |
|---|---|---|---|
| **Training objective** | | **Training data** | |
| Image-Text (Img-Txt) | 14 | ImageNet-1k (IN1k) | 37 |
| Self-Supervised (SSL) | 20 | ImageNet-21k (IN21k) | 9 |
| Supervised (Sup) | 30 | Large | 11 |
| | | XLarge | 7 |
| **Architecture Class** | | **Model Size** | |
| Convolutional (CNN) | 24 | small $< 100M$ parameter | 32 |
| Transformer (TX) | 40 | medium $< 200M$ parameter | 14 |
| | | large $< 400M$ parameter | 8 |
| | | xlarge $> 400M$ parameter | 10 |

**Preprocessing**. To extract the latent representations for each model-dataset combination, we used the Python package

`thingsvision` (Muttenthaler & Hebart, 2021). Images were resized to 256px and center-cropped to 224px before applying the model-specific normalizations from the pretraining. Tab. 2 specifies exactly which layers were used for each model. For ViT models, we used the `CLS` token representations after the final `LayerNorm` module. We applied L2-normalization to ensure all feature vectors are of unit length.

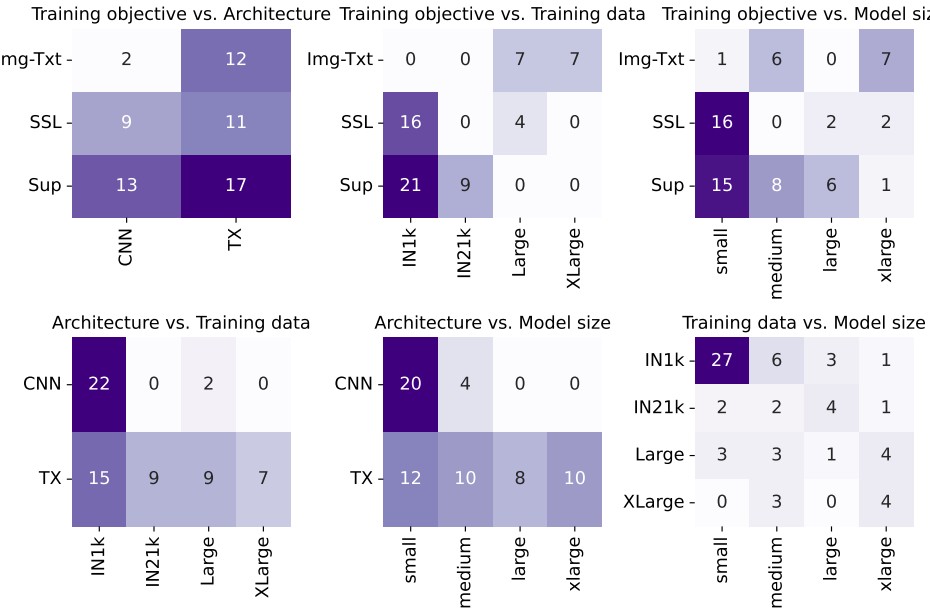

Figure 9: Distribution of models across paired categories. Each heatmap shows the count of vision models for different combinations of model categories: training objective (Sup: supervised, SSL: self-supervised learning, Img-Txt: image-text), architecture (CNN: convolutional neural networks, TX: transformers), training data (IN1k: ImageNet-1k, IN21k: ImageNet-21k, Large/XLarge: larger datasets), and model size (small to xlarge)

## B. Downstream task performance

We evaluated each model's downstream task performance on every dataset by calculating the top-1 accuracy of a linear probe trained on features extracted from the corresponding training set. Each model was evaluated with 3 different random seeds on each dataset, and the mean top-1 accuracy across seeds was used. Hyperparameter selection was performed with a validation set of 20% of the training set and the remaining training data for optimization. We followed the binary search procedure described in (Radford et al., 2021) and searched for the optimal weight decay parameter $\lambda$ in the interval between $10^{-6}$ and $10^2$ in 96 logarithmically spaced steps. This was done for all learning rates $\eta \in \{10^i\}_{i=1}^4$. After hyperparameter selection, the linear probe was retrained on the full training set and evaluated on the respective test set (validation set for Imagenet-1k). All linear probes are trained for 20 epochs, using the AdamW optimizer (Loshchilov & Hutter, 2019) and a cosine schedule for learning rate decay (Loshchilov & Hutter, 2017).

## C. CKA sensitivity to the number of samples in dataset

Our analysis of model similarities requires reliable CKA measurements while managing computational constraints. To identify the minimum number of samples needed for stable CKA values, we analyzed ImageNet-1k subsets with varying samples per class ($k \in 1, 5, 10, 20, 30, 40$). We generated pairwise similarity matrices (Fig. 10) and assessed stability by comparing the absolute differences between consecutive sample sizes (Fig. 11). The results show that most CKA variants converged with 10,000 samples (k=10), with only CKA RBF ($\sigma = 0.2$) requiring more samples (30,000, k=30) for stability.

To further assess CKA stability across different samples, we conducted bootstrapping experiments using six representative models (OpenCLIP ViT-L, OpenCLIP RN50, ResNet-50, ViT-L, DINOv2 ViT-L, and SimCLR RN50—the same anchor models described in Appx. G). For CKA linear, we performed 500 bootstrap iterations with 10,000 images per subset. For CKA RBF ($\sigma = 0.2$), we performed 500 bootstrap iterations with 30,000 images per subset. For each bootstrap sample, we

extracted features from all six models and computed CKA values between each model pair.

Fig. 12(a) confirms the stability of CKA linear values across all model pairs, while for CKA RBF ($\sigma = 0.2$), Fig. 12(b) reveals that even when using the larger 30,000 samples per subset, we still observe slightly higher variability compared to CKA linear.

These findings indicate that local similarity measures (RBF kernel) are more sensitive to specific stimuli than global similarity measures (linear kernel). This aligns with theoretical expectations, as stimulus-specific fine-grained details have a greater influence on local similarity measurements.

Based on these findings, we applied stratified subsampling to limit all datasets to 10,000 or 30,000 samples for CKA linear and CKA RBF ($\sigma = 0.2$), respectively.

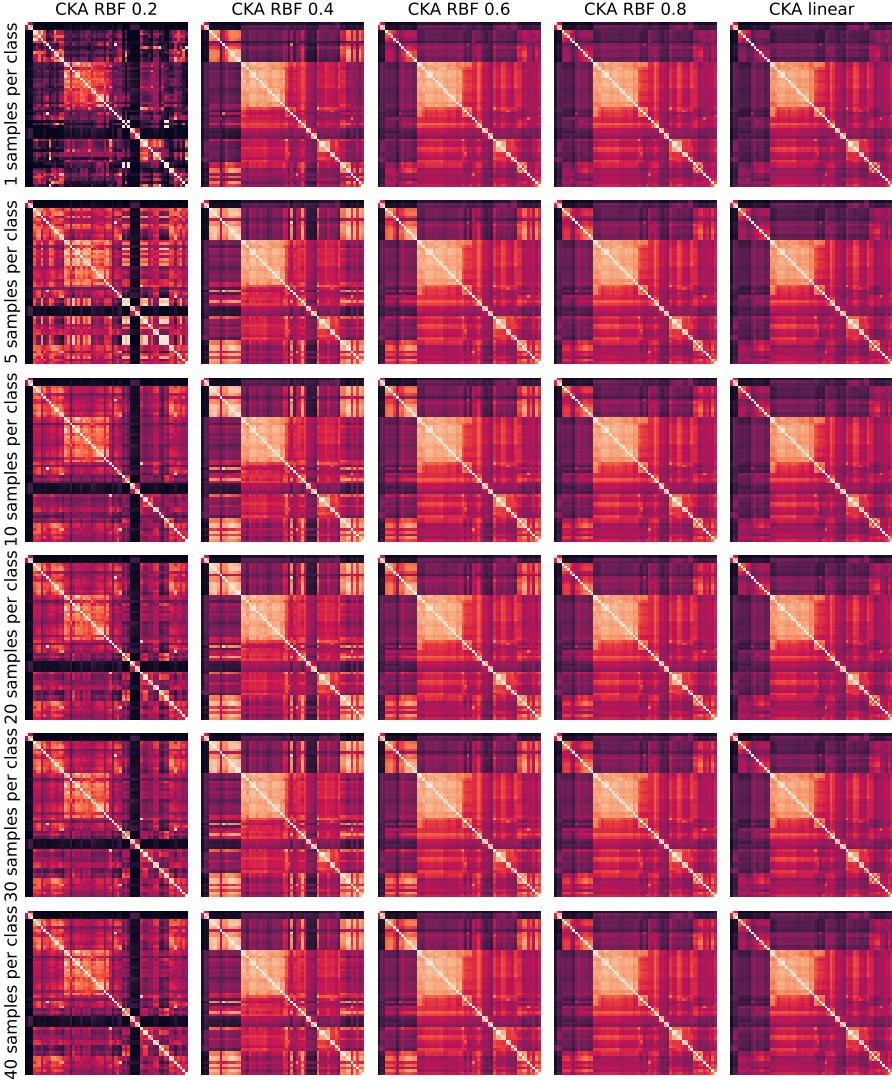

Figure 10: Pairwise model representational similarity matrices using different dataset sizes (1-40 samples per ImageNet-1k class) and metrics (CKA RBF 0.2-0.8 to linear).

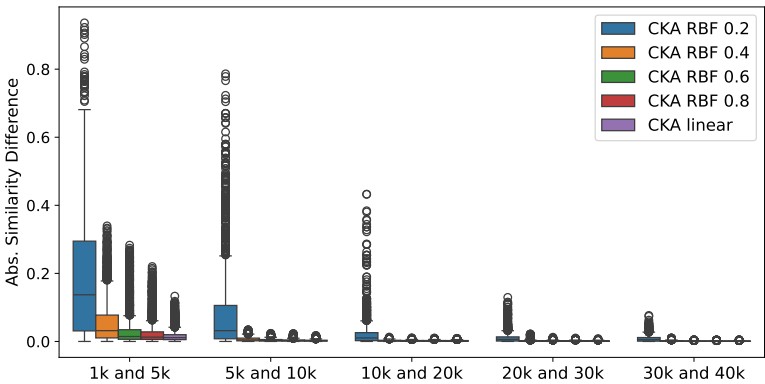

Figure 11: Distribution of absolute differences of pairwise model representational similarities for each CKA variant measured on subsets of ImageNet-1k with consecutive sample sizes per class ($1\rightarrow5, \cdots, 30\rightarrow40$). Most CKA variants converge at $k = 10$ samples per class, with CKA RBF ($\sigma = 0.2$) requiring larger sample sizes for stability.

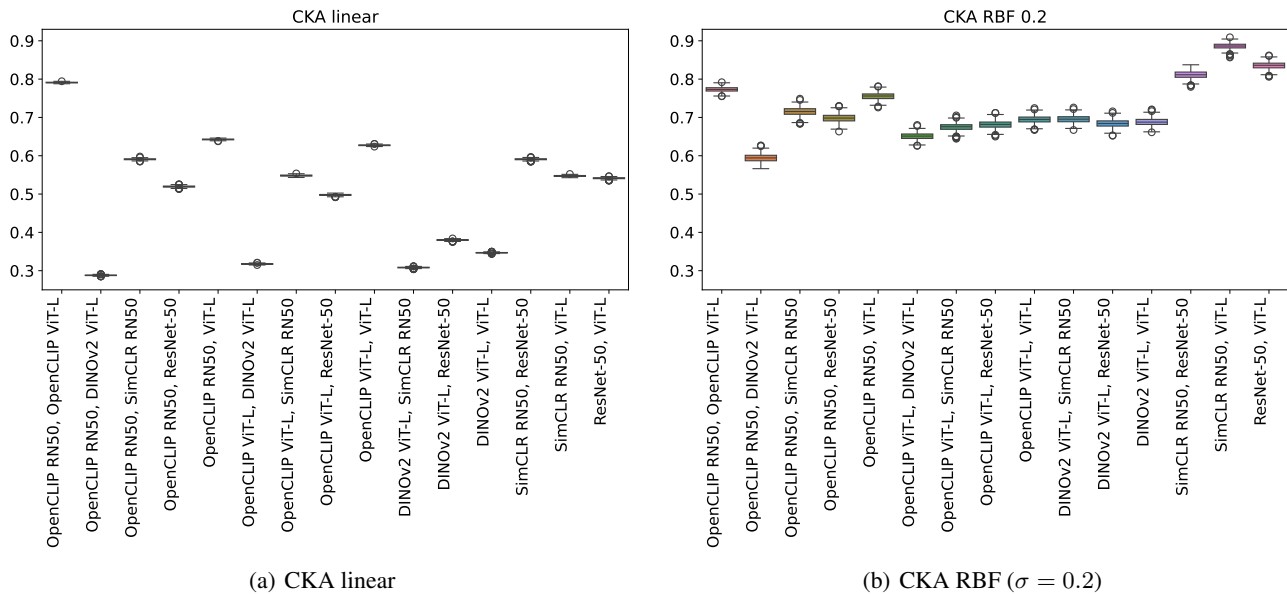

(a) CKA linear

(b) CKA RBF ($\sigma = 0.2$)

Figure 12: Distribution of CKA values between model representations across 500 randomly sampled ImageNet-1k subsets for different model pairs. (a) Linear CKA computed on 10k samples per subset. (b) RBF CKA ($\sigma = 0.2$) computed on 30k samples per subset.

## D. Similarity matrices for different datasets

In § 4.2, we showed that similarity measurements are not directly transferable across datasets due to their high variability. Fig. 2 displays representational similarity matrices computed using CKA linear across three selected datasets, along with the mean and standard deviation matrices across all evaluation datasets. In Fig. 13, we show additional similarity matrices for three representative datasets from each dataset category (see Appx. A for details). The yellow-boxed models, containing image-text models, are most similar on natural image datasets. In contrast, the white-boxed models, containing self-supervised models, show the highest similarity on specialized EuroSAT and PCAM datasets. The cyan-boxed models, containing supervised models, are most dataset dependent, showing high similarity on Pets but low similarity on ImageNet-1k.

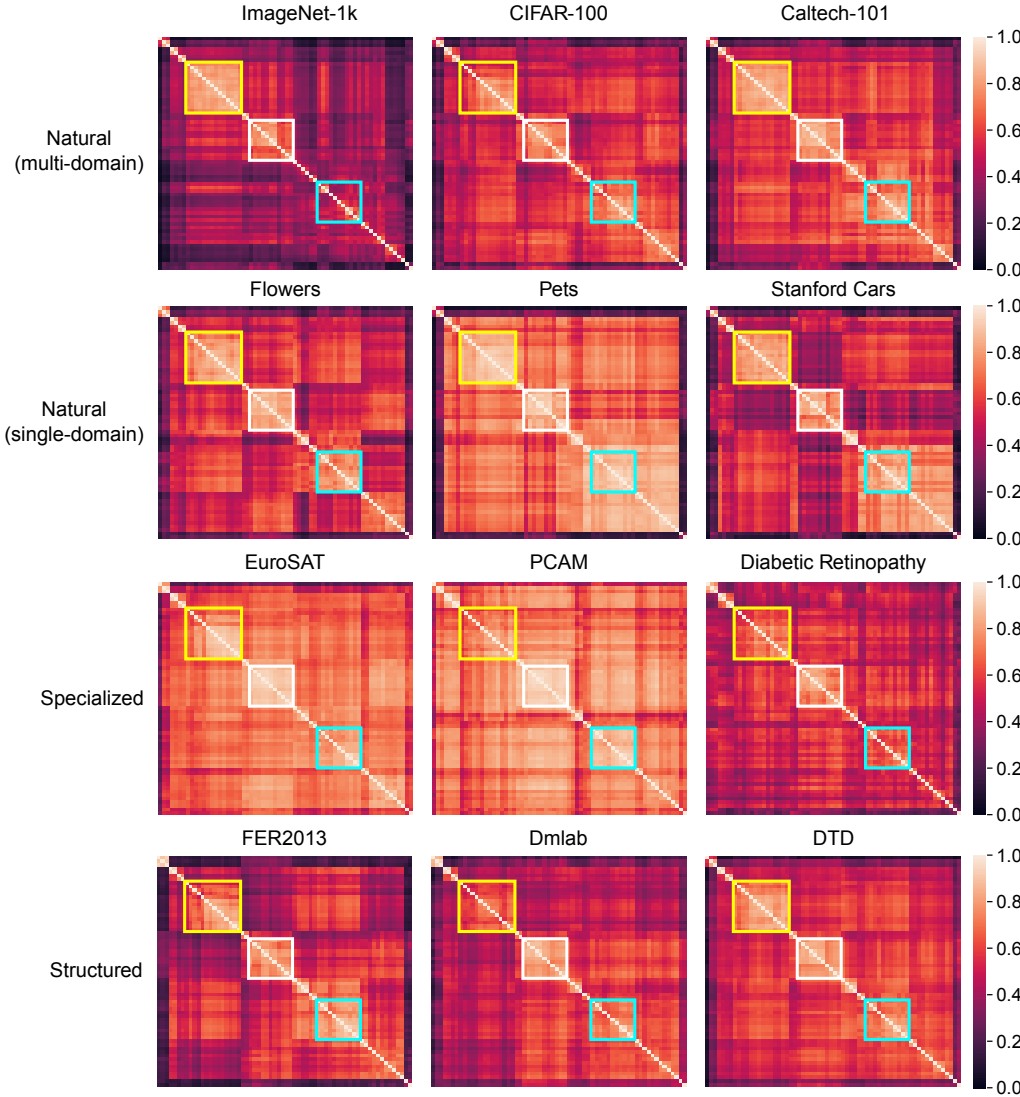

Figure 13: Representational similarity of model pairs, calculated with linear CKA. For each dataset category, three evaluation datasets are displayed.

## E. Variation of CKA values near the upper bound

The CKA similarity measure is bounded between 0 and 1, with values close to 1 indicating highly similar representations. However, this upper bound might lead to saturation effects. To better differentiate between highly similar model representations, we apply two nonlinear transformations: `arccos` and `tan`. Both transformations expand the range of high CKA

values, with `tan` slightly reducing variation for low similarity values. These transformations amplify the inverse U-shape relationship noted in Section 4.2 (Fig. 14), suggesting that the relationship between mean and variance of similarities is robust across different scales.

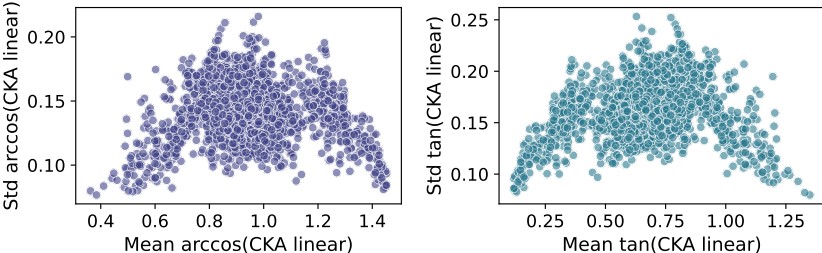

Figure 14: Mean versus standard deviation of CKA values after nonlinear transformations. **Left**: `arccos` transformation inverts the scale, mapping high similarities to low values and vice versa. **Right**: `tan` transformation. Both transformations make the inverse U-shape relationship more pronounced.

## F. Influence of similarity metric on similarity consistency

In our main analysis, we used CKA linear as a global similarity metric. Fig. 3 shows strong positive correlations between mean representational similarities across datasets when measured with different metrics. Here, we investigate whether this stability extends to similarity consistency distributions when using alternative metrics.

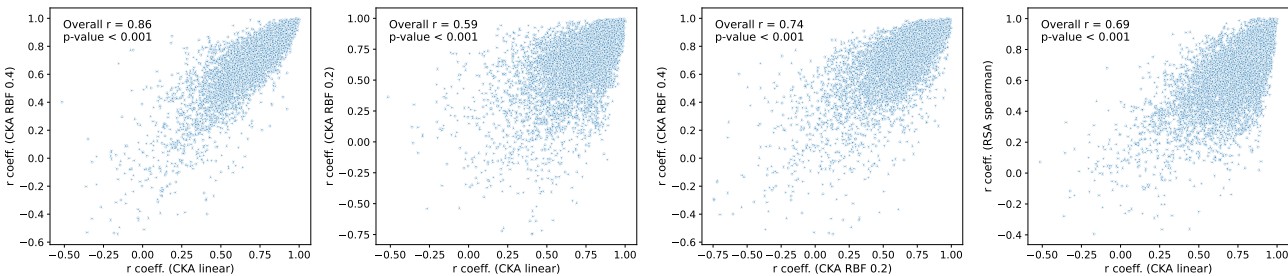

Figure 15: Relationship between similarity consistency values measured with different similarity metrics across all model sets and dataset pairs. From left to right: CKA linear vs. CKA RBF ($\sigma = 0.4$), CKA linear vs. CKA RBF ($\sigma = 0.2$), CKA RBF ($\sigma = 0.2$) vs. CKA RBF ($\sigma = 0.4$), and CKA linear vs. RSA Spearman. Correlations are measured using Pearson correlation. Each point represents the similarity consistency value for a pair of datasets and model sets.

**Local similarity metrics**  When examining local similarity structures, we leverage the flexibility of CKA with an RBF kernel. The kernel bandwidth parameter $\sigma$ controls sensitivity to local structure, allowing us to capture different aspects of representational similarity. Notably, Fig. 10 reveals that pattern differences between CKA linear and CKA RBF ($\sigma = 0.2$) are substantially more pronounced than those between CKA linear and CKA RBF ($\sigma = 0.4$). This suggests that the latter partially captures global similarity structures rather than purely local ones. Indeed, the correlation between similarity consistency values measured with CKA linear and CKA RBF ($\sigma = 0.4$) is strong (Fig. 15, leftmost panel), while it is substantially lower between CKA linear vs. CKA RBF ($\sigma = 0.2$) (Fig. 15, second panel). The correlation between similarity consistency values measured with the two different sigmas ($\sigma = 0.2$ and $\sigma = 0.4$) (Fig. 15, third panel) is in between, strengthening the hypothesis that CKA RBF ($\sigma = 0.4$) partially captures global structure. Therefore, we use CKA RBF ($\sigma = 0.2$) as our local similarity metric.

We observe the same overall pattern as in Fig. 6 of the main text for the CKA RBF ($\sigma = 0.2$) similarity metric: the objective is a main influencing factor for similarity consistency, while network architecture and model size are less important (Fig. 16). However, the distributions of consistency values reveal two interesting observations. For local similarity, supervised model pairs ($\Phi_{\text{Sup}}$, $\Phi_{\text{Sup}}$) are more consistent than the image-text model pairs ($\Phi_{\text{Img-Txt}}$, $\Phi_{\text{Img-Txt}}$), which are better at representing global structure. In addition, models trained on IN21k ($\Phi_{\text{IN21k}}$, $\Phi_{\text{IN21k}}$) are more consistent than models trained on IN1k

($\Phi_{IN1k}$, $\Phi_{IN1k}$). Their ordering flipped compared to Fig. 6, where ($\Phi_{IN1k}$, $\Phi_{IN1k}$) model pairs are more consistent in their global structure. IN21k contains substantially more classes (21.843) and a higher percentage of the classes are leaf nodes in the WordNet (Princeton University, 2010) tree (76.71% for IN21k vs. 65% for IN1k), representing more fine-grained entities. The representation must contain more fine-grained details to distinguish these classes, dominating local similarities.

**Other global similarity metrics** In Fig. 17, the correlation between the two global measures (CKA linear and RSA Spearman) is moderate. Even with this reduced correlation strength, similarity consistency measured using RSA Spearman exhibits the same patterns as CKA linear, particularly regarding the model category we found to be most influential – the training objective (see Fig. 17).

The analysis presented in this section suggests that our observations on the influence of the training objective on similarity consistency hold for multiple metrics.

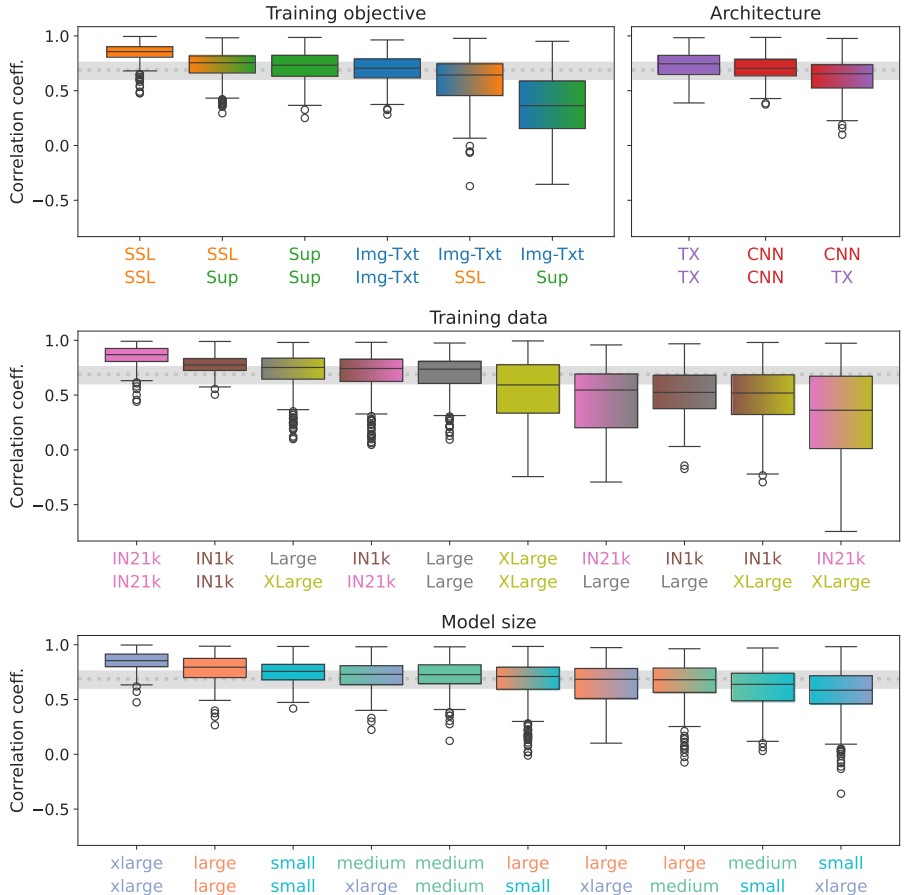

Figure 16: Distribution of similarity consistencies for each model set pair based on CKA RBF ($\sigma = 0.2$) representational similarities. The boxes are sorted in decreasing median correlation. The dotted line indicates the overall median, while the gray area spans the 25th to 75th percentiles of correlations across all model pairs.

## G. Validating training objective effects through controlled anchor model analysis

To validate our findings regarding the influence of different model categories (§4.5), we conducted a controlled analysis using six carefully selected anchor models: OpenCLIP RN50, OpenCLIP ViT-L, SimCLR RN50, DINOv2 ViT-L, ResNet-50, and ViT-L. These models were chosen to create systematic variations across training objectives (image-text, self-supervised, and supervised learning), while controlling for architecture (convolutional ResNet-50 vs transformer ViT-L) and model size. This controlled setup helps verify whether the patterns observed in our main analysis persist when comparing individual representative models to broader model sets. For each anchor model, we define a single-element model set (e.g.,

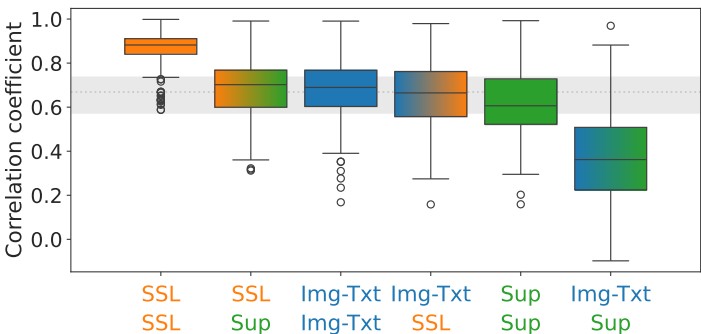

Figure 17: Distribution of similarity consistency values for each model set pair within the training objective category, based on RSA Spearman representational similarities

$\Phi_{\text{OpenCLIP RN50}}$) and compute its similarity consistency with different model groups (e.g., $R(\Phi_{\text{OpenCLIP RN50}}, \Phi_{\text{SSL}})$).

Fig. 18 shows the correlation coefficient distributions across dataset pairs, focusing on training objectives as they emerged as the most influential factor for similarity consistency. The results support our main findings while providing additional insights. Self-supervised models (SimCLR RN50, DINOv2 ViT-L) consistently show the highest median correlations and lowest variations when compared with any anchor model, regardless of the anchor's training objective. Surprisingly, for global similarity, even image-text (OpenCLIP RN50, OpenCLIP ViT-L) and supervised (ResNet-50, ViT-L) anchor models correlate more strongly with $\Phi_{\text{SSL}}$ than with their own categories ($\Phi_{\text{Img-Txt}}$ and $\Phi_{\text{Sup}}$, respectively). However, the image-text anchors show high variation of similarity consistency for local similarity measures and all training objectives. We also observe a notable pattern of weak correlations between image-text anchors and supervised models ($\Phi_{\text{Sup}}$) and vice versa. These patterns remain remarkably consistent across different architectures and model sizes, suggesting that training objectives, rather than architectural choices or model sizes, primarily drive similarity relationships across datasets.

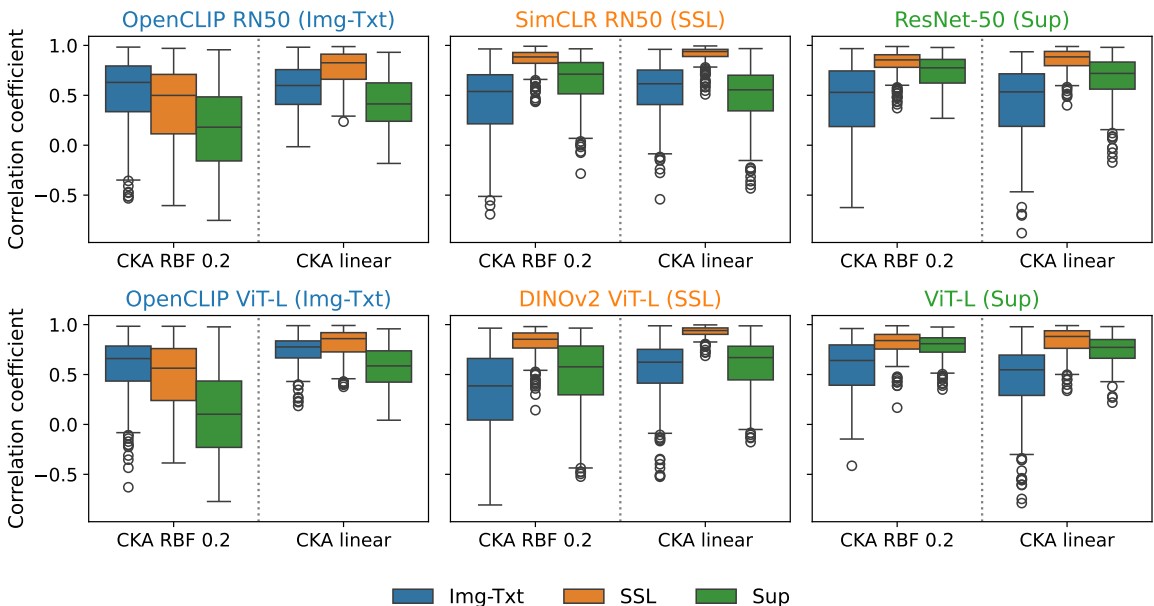

Figure 18: Distribution of Pearson correlation coefficients between anchor models (OpenCLIP RN50, OpenCLIP ViT-L, SimCLR RN50, DINOv2 ViT-L, ResNet-50, ViT-L) and models trained with different objectives. Each box shows correlations across all dataset pairs between one anchor model and models from specific sets (e.g., $\Phi_{\text{SSL}}$, $\Phi_{\text{Sup}}$). Results for CKA RBF ($\sigma = 0.2$) and CKA linear are separated by dotted lines. Subplot titles are colored according to the anchor model's subcategory.

## H. Effect of evaluation data on similarity consistency

In the main analysis, we examined similarity consistency across model categories for all evaluation datasets (§4.5) and across individual datasets for all models (§4.6). Here, we investigate whether model category effects differ between natural and non-natural (specialized and structured) datasets, as non-natural datasets represent out-of-distribution data for many models (Fig. 19).

The primary finding from §4.5 holds: training objective remains the dominant factor influencing similarity consistency regardless of dataset type. However, similarity consistency between image-text and supervised models ($\Phi_{\text{Img-Txt}}$, $\Phi_{\text{Sup}}$) shows lower medians and higher variance on natural datasets compared to non-natural ones. For natural datasets, supervised models might encode features tightly coupled to class-label supervision, which diverge from the semantically richer, language-aligned representations of image-text models. Conversely, on non-natural datasets where both model types operate further from their training distributions, this representational gap narrows as both might rely on more generalized features. Complementary, the analysis—examining how training dataset composition affects these patterns—is presented in the next section.

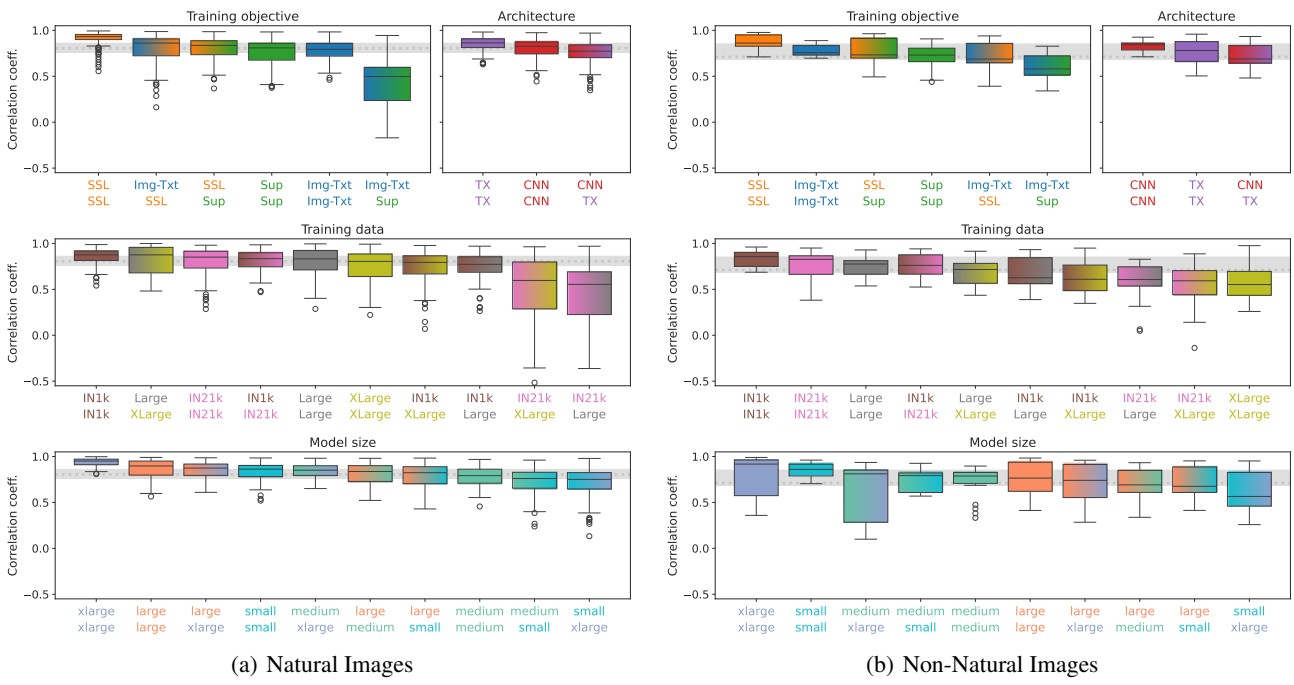

(a) Natural Images        (b) Non-Natural Images

Figure 19: Distribution of similarity consistencies for natural (single- and multi-domain) and non-natural (specialized and structured) datasets. Representational similarities for each model set pair are computed with CKA linear.

## I. Effect of training data domain on similarity consistency

In our main analysis, we restricted our focus to general-purpose vision models trained on large-scale datasets with diverse semantic classes (§ 4.1). This focus was motivated by the Platonic representation hypothesis, which suggests that as the semantic diversity of training data approaches the complexity of reality itself, representations should increasingly converge (Huh et al., 2024). However, Conwell et al. (2024) demonstrated that training data is a key driver of representational similarity between models, potentially more significant than architecture, training objective, or choice of similarity metric. In this section, we examine how this observation manifests in terms of representational consistency.

To investigate the role of training data domain, we selected four architectures (AlexNet (Krizhevsky et al., 2012a), DenseNet161 (Huang et al., 2017), ResNet18 (He et al., 2016), and ResNet50 (He et al., 2016)) with available pre-trained weights for both the general-purpose ImageNet-1k and the scene-specific Places365 (Zhou et al., 2018) datasets. Models

were sourced from torchvision[1], using default ImageNet weights and Places365 weights from the official GitHub repository[2]. This selection of models allowed us to isolate the training data effect on similarity consistency while fixing the other characteristics: training objective (Supervised), architecture type (CNN), and model size (small).

We evaluated similarity consistency across the 23 downstream datasets for all model set pairs: $(\Phi_{\text{IN1k}}, \Phi_{\text{IN1k}})$, $(\Phi_{\text{IN1k}}, \Phi_{\text{Places365}})$, $(\Phi_{\text{Places365}}, \Phi_{\text{Places365}})$, and $(\Phi_{\text{all}}, \Phi_{\text{all}})$, where $\Phi_{\text{all}}$ contains all models.

Our analysis reveals two main observations. First, as shown in Fig. 20, Places365-trained model pairs demonstrate more consistent representational similarities compared to ImageNet-1k-trained pairs We hypothesize that domain-specific training constrains the solution space more tightly, whereas the greater diversity of semantic classes in ImageNet-1k allows for more varied learned representations across models. This distinction is particularly interesting in light of the platonic representation hypothesis. While domain-specific models show higher consistency, this may reflect convergence to specialized solutions rather than to general representations of reality. In contrast, the lower consistency among ImageNet-1k-trained models likely reflects the challenge of learning representations that capture broad semantic diversity. This consistency might be even lower for models trained with supervised objectives since they can exploit different confounders and shortcuts when learning to classify the diverse set of classes.

Second, the distributions $R(\Phi_{\text{IN1k}}, \Phi_{\text{IN1k}})$, $R(\Phi_{\text{IN1k}}, \Phi_{\text{Places365}})$, and $R(\Phi_{\text{all}}, \Phi_{\text{all}})$ exhibit similar distribution spreads. Additionally, $R(\Phi_{\text{all}}, \Phi_{\text{all}})$ shows a distribution similar to $R(\Phi_{\text{IN1k}}, \Phi_{\text{IN1k}})$ in Fig. 6. This suggests that including domain-specific models alongside general-purpose ones might not fundamentally alter the representational similarity consistency patterns observed in our analysis.

Our analysis shows that the training data domain significantly influences representational similarity consistency, though mixing domain-specific and general-purpose models preserves the broad distribution patterns of correlation coefficients observed in ImageNet-1k models alone.

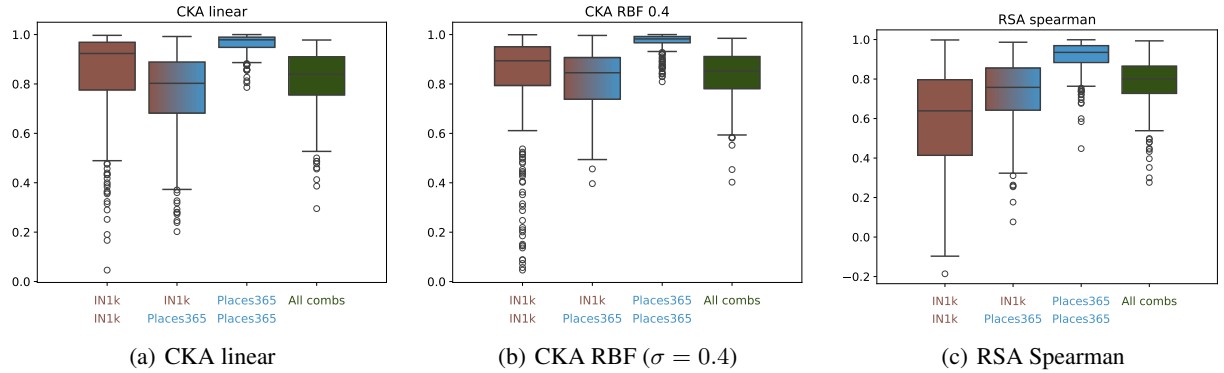

(a) CKA linear          (b) CKA RBF ($\sigma = 0.4$)          (c) RSA Spearman

Figure 20: Distribution of Pearson correlation coefficients for model similarities computed with different similarity metrics across dataset pairs, i.e., $R(\Phi_{\text{IN1k}}, \Phi_{\text{IN1k}})$, $R(\Phi_{\text{IN1k}}, \Phi_{\text{Places365}})$, $R(\Phi_{\text{Places365}}, \Phi_{\text{Places365}})$, and $R(\Phi_{\text{all}}, \Phi_{\text{all}})$, with $\Phi_{\text{IN1k}}$ containing the four IN1k and $\Phi_{\text{Places365}}$ containing the four Places365 pre-trained models. $\Phi_{\text{all}}$ contains all eight models.

## J. Distribution correlation coefficients downstream task performance vs. model similarity

In §4.7, we examined the correlations between representational similarity and classification performance gaps across three datasets from each of the four dataset categories: Natural (multi-domain), Natural (single-domain), Specialized, and Structured. We observed stronger negative correlations for the Natural (single-domain) datasets. Fig. 22 extends this analysis by illustrating the correlation coefficients for all evaluated datasets. We can confirm that natural (single-domain) datasets exhibit the strongest negative correlations overall on this larger set of datasets, with the SVHN dataset being the sole outlier.

Interestingly, ImageNet-1k exhibits the weakest negative correlation between performance gap and representational similarity.

---

[1] https://pytorch.org/vision/stable/index.html
[2] https://github.com/CSAILVision/places365

We hypothesize that this is due in part to the high diversity of classes in ImageNet-1k, along with its rich set of confounders and contextual cues, which enable networks to approach the classification task in multiple ways. This may be exacerbated by the fact that some of the evaluated networks were pretrained on ImageNet-1k or ImageNet-21k, which could have encouraged the learning of dataset-specific features that perform well on ImageNet but fail to generalize to other datasets.

While our categorization of natural datasets into single-domain and multi-domain revealed an interesting difference in the relationship between downstream task performance and model similarity—especially between the ImageNet-1k and single-domain datasets like Flowers or Pets—the correlation coefficients for these categories still overlap. We hypothesize that our binary sub-categorization of Natural datasets is insufficient for capturing a clear distinction for two reasons: (1) Datasets lie on a spectrum between the two (hypothetical) extremes of encompassing all possible domains and encompassing exactly one. For instance, ImageNet-1k has a relatively rich set of domains, containing 1,000 diverse classes, whereas CIFAR-10 only contains 10 classes of vehicles and animals. (2) Datasets differ in the amount of contextual information (e.g., the object's background) they offer. We assume that contextual information plays an important role in the number of viable features a dataset classification can be solved with. A large number of viable features means that multiple well-performing but dissimilar strategies may exist. Consequently, multi-domain datasets with limited contextual information may behave more like single-domain datasets. For example, the PASCAL VOC 2007 dataset, when used for single-label classification, makes use of tight bounding-box crops around the object to be classified, thereby reducing the available context. Similarly, in low-resolution datasets such as CIFAR-10 and CIFAR-100, contextual information may be reduced due to the loss of background information. While we have not quantified it, we found the Caltech-101 dataset to contain much cleaner object backgrounds than ImageNet-1k, also suggesting reduced contextual information.

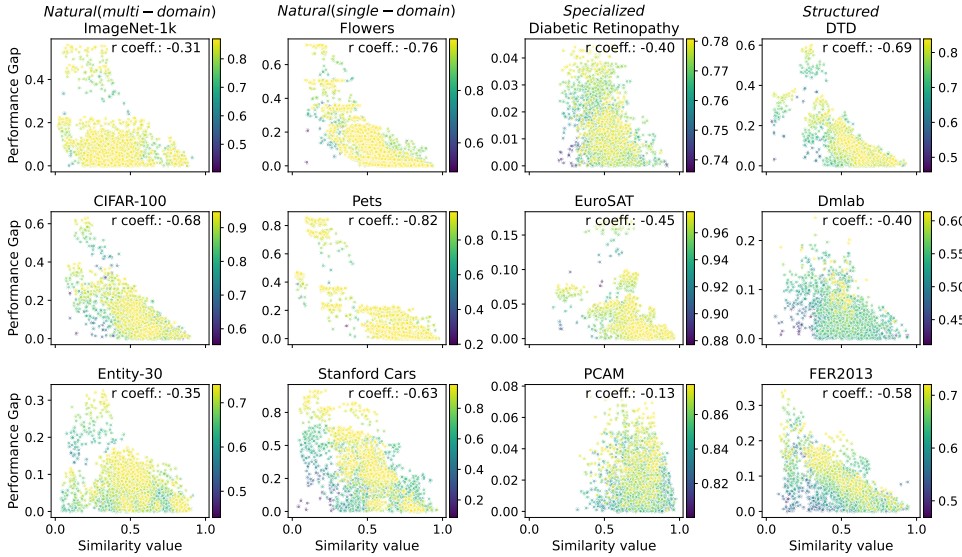

Figure 21: Model similarity (CKA linear) vs. absolute difference in downstream task performance (top-1 accuracy) for model pairs for three datasets per dataset category ( natural multi-domain, natural single-domain, specialized, and structured). The color of each point indicates the downstream task accuracy of the better-performing model. Pearson correlation coefficients are shown in each subplot.

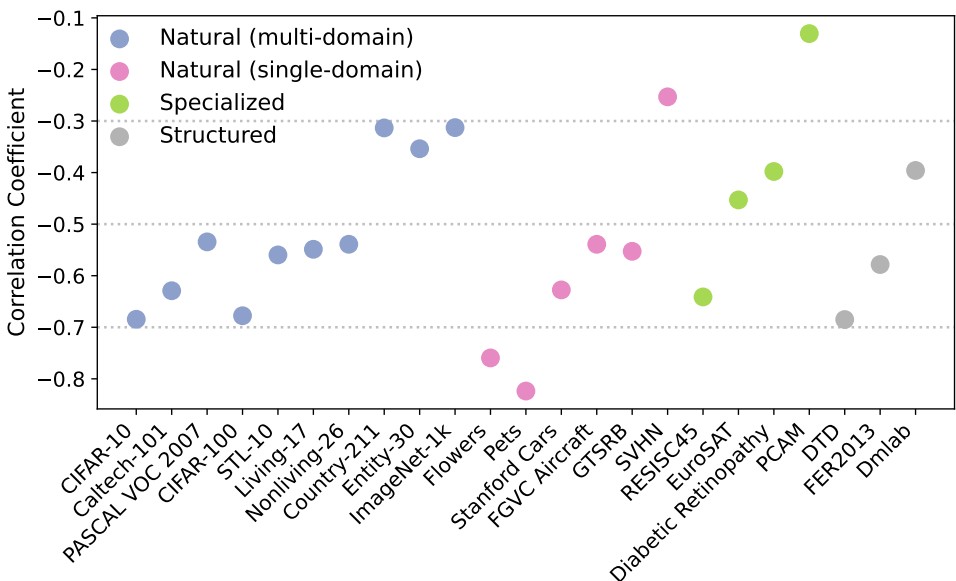

Figure 22: Pearson correlation coefficients between downstream task performance gap and model representation similarity for all 23 datasets grouped by dataset category.

Table 2: Pretrained neural networks that we considered in our analyses.

| Model name | Source | Rep. Layer | Training objective | Training data | Training data class | Architecture | Architecture class | Model size | Model size class |
|---|---|---|---|---|---|---|---|---|---|
| Kakaobrain-Align | KakaoBrain (2023) | pooler | Img-Txt | COYO-700M | Large | EfficientNet | CNN | 62.1M | small |
| OpenCLIP-EVA01-g-14-plus-merged2b-s11b-b114k | Sun et al. (2023) | visual | Img-Txt | Merged2B | XLarge | ViT | TX | 1.4B | xlarge |
| OpenCLIP-EVA01-g-14-laion400m-s11b-b41k | Sun et al. (2023) | visual | Img-Txt | LAION400M | Large | ViT | TX | 1.1B | xlarge |
| OpenCLIP-EVA02-B-16-merged2b-s8b-b131k | Sun et al. (2023) | visual | Img-Txt | Merged2B | XLarge | ViT | TX | 149.7M | medium |
| OpenCLIP-EVA02-L-14-merged2b-s4b-b131k | Sun et al. (2023) | visual | Img-Txt | Merged2B | XLarge | ViT | TX | 427.8M | large |
| OpenCLIP-RN50-openai | Radford et al. (2021) | visual | Img-Txt | WIT-400M | Large | ResNet | CNN | 102.0M | medium |
| OpenCLIP-ViT-B-16-SigLIP-webli | Zhai et al. (2023) | visual | Img-Txt | WebLI | XLarge | ViT | TX | 203.2M | medium |
| OpenCLIP-ViT-B-16-laion2b-s34b-b88k | Radford et al. (2021) | visual | Img-Txt | LAION2B | XLarge | ViT | TX | 149.6M | medium |
| OpenCLIP-ViT-B-16-laion400m-e32 | Radford et al. (2021) | visual | Img-Txt | LAION400M | Large | ViT | TX | 149.6M | medium |
| OpenCLIP-ViT-B-16-openai | Radford et al. (2021) | visual | Img-Txt | WIT-400M | Large | ViT | TX | 149.6M | medium |
| OpenCLIP-ViT-L-14-laion2b-s32b-b82k | Radford et al. (2021) | visual | Img-Txt | LAION2B | XLarge | ViT | TX | 427.6M | xlarge |
| OpenCLIP-ViT-L-14-laion400m-e32 | Radford et al. (2021) | visual | Img-Txt | LAION400M | Large | ViT | TX | 427.6M | xlarge |
| OpenCLIP-ViT-L-14-openai | Radford et al. (2021) | visual | Img-Txt | WIT-400M | Large | ViT | TX | 427.6M | xlarge |
| vit-huge-patch14-clip-224.laion2b | Radford et al. (2021) | norm | Img-Txt | LAION2B | XLarge | ViT | TX | 632.1M | xlarge |
| barlowtwins-rn50 | Zbontar et al. (2021) | avgpool | SSL | IN1k | IN1k | ResNet | CNN | 23.5M | small |
| dino-rn50 | Caron et al. (2021) | avgpool | SSL | IN1k | IN1k | ResNet | CNN | 23.5M | small |
| dino-vit-base-p16 | Caron et al. (2021) | norm | SSL | IN1k | IN1k | ViT | TX | 85.8M | small |
| dino-vit-small-p16 | Caron et al. (2021) | norm | SSL | IN1k | IN1k | ViT | TX | 21.7M | small |
| dino-xcit-medium-24-p16 | Caron et al. (2021) | norm | SSL | IN1k | IN1k | ViT | TX | 83.9M | small |
| dino-xcit-small-12-p16 | Caron et al. (2021) | norm | SSL | IN1k | IN1k | ViT | TX | 25.9M | small |
| dinov2-vit-base-p14 | Oquab et al. (2024) | norm | SSL | LVD-142M | Large | ViT | TX | 86.6M | small |
| dinov2-vit-giant-p14 | Oquab et al. (2024) | norm | SSL | LVD-142M | Large | ViT | TX | 1.1B | xlarge |
| dinov2-vit-large-p14 | Oquab et al. (2024) | norm | SSL | LVD-142M | Large | ViT | TX | 304.4M | large |
| dinov2-vit-small-p14 | Oquab et al. (2024) | norm | SSL | LVD-142M | Large | ViT | TX | 22.1M | small |
| jigsaw-rn50 | Noroozi & Favaro (2016) | avgpool | SSL | IN1k | IN1k | ResNet | CNN | 23.5M | small |
| mae-vit-base-p16 | He et al. (2022) | norm | SSL | IN1k | IN1k | ViT | TX | 86.4M | small |
| mae-vit-huge-p14 | He et al. (2022) | norm | SSL | IN1k | IN1k | ViT | TX | 630.8M | xlarge |
| mae-vit-large-p16 | He et al. (2022) | norm | SSL | IN1k | IN1k | ViT | TX | 303.3M | large |
| mocov2-rn50 | Chen et al. (2020c) | avgpool | SSL | IN1k | IN1k | ResNet | CNN | 23.5M | small |
| pirl-rn50 | Misra & van der Maaten (2020) | avgpool | SSL | IN1k | IN1k | ResNet | CNN | 23.5M | small |
| rotnet-rn50 | Gidaris et al. (2018) | avgpool | SSL | IN1k | IN1k | ResNet | CNN | 23.5M | small |
| simclr-rn50 | Chen et al. (2020a) | avgpool | SSL | IN1k | IN1k | ResNet | CNN | 23.5M | small |
| swav-rn50 | Caron et al. (2020) | avgpool | SSL | IN1k | IN1k | ResNet | CNN | 23.5M | small |
| vicreg-rn50 | Bardes et al. (2022) | avgpool | SSL | IN1k | IN1k | ResNet | CNN | 23.5M | small |
| beit-base-patch16-224 | Bao et al. (2022) | norm | Sup | IN21k + IN1k | IN21k | ViT | TX | 86.5M | small |
| beit-base-patch16-224.in22k-ft-in22k | Bao et al. (2022) | norm | Sup | IN21k | IN21k | ViT | TX | 102.6M | medium |
| beit-large-patch16-224 | Bao et al. (2022) | norm | Sup | IN21k + IN1k | IN21k | ViT | TX | 304.4M | large |
| beit-large-patch16-224.in22k-ft-in22k | Bao et al. (2022) | norm | Sup | IN21k | IN21k | ViT | TX | 325.8M | large |
| convnext-base | Liu et al. (2022) | head.flatten | Sup | IN1k | IN1k | ConvNeXt | CNN | 88.6M | small |
| convnext-large | Liu et al. (2022) | head.flatten | Sup | IN1k | IN1k | ConvNeXt | CNN | 197.8M | medium |
| deit3-base-patch16-224 | Touvron et al. (2021) | norm | Sup | IN1k | IN1k | ViT | TX | 86.6M | small |
| deit3-base-patch16-224.fb-in22k-ft-in1k | Touvron et al. (2021) | norm | Sup | IN21k + IN1k | IN21k | ViT | TX | 86.6M | small |
| deit3-large-patch16-224 | Touvron et al. (2021) | norm | Sup | IN1k | IN1k | ViT | TX | 304.4M | large |
| deit3-large-patch16-224.fb-in22k-ft-in1k | Touvron et al. (2021) | norm | Sup | IN21k + IN1k | IN21k | ViT | TX | 304.4M | large |
| efficientnet-b3 | Tan & Le (2019) | avgpool | Sup | IN1k | IN1k | EfficientNet | CNN | 12.2M | small |
| efficientnet-b4 | Tan & Le (2019) | avgpool | Sup | IN1k | IN1k | EfficientNet | CNN | 19.3M | small |
| efficientnet-b5 | Tan & Le (2019) | avgpool | Sup | IN1k | IN1k | EfficientNet | CNN | 30.4M | small |
| efficientnet-b6 | Tan & Le (2019) | avgpool | Sup | IN1k | IN1k | EfficientNet | CNN | 43.0M | small |
| efficientnet-b7 | Tan & Le (2019) | avgpool | Sup | IN1k | IN1k | EfficientNet | CNN | 66.3M | small |
| resnet152 | He et al. (2016) | avgpool | Sup | IN1k | IN1k | ResNet | CNN | 60.2M | small |
| resnet50 | He et al. (2016) | avgpool | Sup | IN1k | IN1k | ResNet | CNN | 25.6M | small |
| resnext50-32x4d | He et al. (2016) | global_pool | Sup | IN1k | IN1k | ResNeXt | CNN | 25.0M | small |
| seresnet50 | Hu et al. (2018) | global_pool | Sup | IN1k | IN1k | SE-ResNet | CNN | 28.1M | small |
| swin-base-patch4-window7-224 | Liu et al. (2021) | global_pool | Sup | IN1k | IN1k | Swin-Transformer | TX | 87.8M | small |
| swin-base-patch4-window7-224.ms-in22k | Liu et al. (2021) | global_pool | Sup | IN21k | IN21k | Swin-Transformer | TX | 109.1M | medium |
| swin-large-patch4-window7-224 | Liu et al. (2021) | global_pool | Sup | IN1k | IN1k | Swin-Transformer | TX | 196.5M | medium |
| swin-large-patch4-window7-224.ms-in22k | Liu et al. (2021) | global_pool | Sup | IN21k | IN21k | Swin-Transformer | TX | 228.6M | medium |
| vgg16 | Simonyan & Zisserman (2015) | classifier.3 | Sup | IN1k | IN1k | VGG | CNN | 138.4M | medium |
| vgg19 | Simonyan & Zisserman (2015) | classifier.3 | Sup | IN1k | IN1k | VGG | CNN | 143.7M | medium |
| vit-base-patch16-224 | Dosovitskiy et al. (2021) | norm | Sup | IN1k | IN1k | ViT | TX | 86.6M | small |
| vit-base-patch16-224.augreg-in21k | Dosovitskiy et al. (2021) | norm | Sup | IN21k | IN21k | ViT | TX | 102.6M | medium |
| vit-huge-patch14-224.orig-in21k | Dosovitskiy et al. (2021) | norm | Sup | IN21k | IN21k | ViT | TX | 630.8M | xlarge |
| vit-large-patch16-224 | Dosovitskiy et al. (2021) | norm | Sup | IN1k | IN1k | ViT | TX | 304.3M | large |
| vit-large-patch16-224.augreg-in21k | Dosovitskiy et al. (2021) | norm | Sup | IN21k | IN21k | ViT | TX | 325.7M | large |

