# OpenReview forum: "Objective drives the consistency of representational similarity across datasets"
_ICML.cc/2025/Conference — ICML 2025 poster_

### Official Review · Reviewer_9H2r · 2025-03-11

**Overall Recommendation:** 4

**Summary:**

To compare representation spaces through representational similarity analysis (RSA) or its close relative in machine learning, centered kernel alignment (CKA), a sample of data is embedded in two different spaces, and the pairwise similarities of all representations in each space is used as a fingerprint for its information content.  The CKA or RSA value critically depends on the dataset from which the sample was drawn.

The current submission looks at the correlation of CKA values across datasets to draw conclusions about the similarities of vision models, or vice versa (correlation of CKA across models for conclusions about datasets), with motivation in large part to check the recent Platonic representation hypothesis (i.e. that the representation spaces of foundation models are converging).

## Update after rebuttal
The authors have addressed all of my main concerns; I appreciate the clarity of the rebuttal and the additional experiments to test the hypotheses I raised.  There are interesting results and the analysis is sound.  I have adjusted my score from 3->4.

**Claims And Evidence:**

The claims are descriptive of the correlations the authors find and are supported by the results.

**Essential References Not Discussed:**

None to my awareness.

**Experimental Designs Or Analyses:**

There are relatively few design choices in the work.  One issue I see relates to the selection of a length scale $\sigma=0.4$ for CKA RBF.  Can the authors justify why using a single value across the board is reasonable?  This assumes that a length of 0.4 has the same relevance in all of the models and for all of the datasets.  Why not use something adaptive, like the median distance between points?  I also did not find the ablation on this parameter (Fig 9) to support CKA RBF with $\sigma=0.4$ as a local probe -- only 0.2 looks like it captures anything substantially different from CKA linear.

**Methods And Evaluation Criteria:**

The first major analysis aggregates across ~20 datasets to compare models, but the rationale for this aggregation warrants further scrutiny.  Since the datasets are an arbitrary selection (not necessarily representative of any relevant distribution of images, with ~5% diabetic retinopathy images) and are weighted evenly in the correlation values, the resulting metrics might reflect the choice of datasets rather than similarities between the models.  If, for example, the proportion of datasets whose domain differs significantly from natural images were 10% or 50% instead of the ~25% used in the work, the correlation values could be entirely different.

To phrase it differently, while CKA values can be related for any manner of producing different samples, the particular dataset selection appears likely to be a major driver in the observed trends.  In order to interpret the model similarity results (aggregating across datasets), it might be important to investigate the effect of such dataset selection.

Aggregating across models to compare datasets seems less problematic -- largely because there does not seem to be the same sense of outliers in the selection of models as there are for datasets.

**Other Comments Or Suggestions:**

- I think the boxes in Fig 2 actually make it harder to assess similarity structure across models.  Perhaps signify at the boundary of the heatmap?

- Might a bootstrap-type analysis act to ground the CKA values for each model and each dataset?  To be specific, one could compute CKA for different samples (size 10k, as is currently used in the submission) from each dataset to get a spread of values, and this spread would help shed light on what constitutes a meaningful difference in CKA values when comparing across models/datasets.

**Other Strengths And Weaknesses:**

The primary contribution relates to the large-scale analysis of 64 models and 23 datasets, as the methodological innovation is limited: essentially performing correlation of CKA values.  As such, the contribution is somewhat slim, and the insights obtained are not particularly actionable.  Still, the paper is well-written and the results are presented clearly.

**Questions For Authors:**

.

**Relation To Broader Scientific Literature:**

The submission can be seen as a direct response to the Platonic representation hypothesis (Huh et al 2024), where the dependence of CKA on dataset immediately raises the question about how dataset factors into the PRH.  While there have been many works on assessing representational similarity, the contributions of this work are not in this direction: it primarily adopts one (the linear variant of CKA) and runs that through various levels of analysis.

**Theoretical Claims:**

I did not see any theoretical claims.

---

> ### Author Rebuttal · Authors · 2025-03-31
>
> We thank the reviewer for acknowledging that _the paper is well-written_ and that our claims are _supported by the results_, which _are presented clearly_. We are grateful for the valuable feedback that helped us improve our paper. We will address each concern point by point. All new figures/tables (R4F1-F7, R4T1-T2) are available [here](https://anonymous.4open.science/r/rebuttal_similarity_consistency/README.md).
>
> First, we want to address _[...] metrics might reflect the choice of datasets rather than similarities between the models_. Our analysis included 23 datasets, mostly part of the VTAB [1] and commonly used in CV. We compute the correlation of similarity values for all pairwise combinations of datasets. Therefore, we obtain similarity consistency measures for all dataset combinations (e.g., natural vs natural, structured vs structured, natural vs structured, …). These are responsible for the variance shown in our boxplot (Fig. 5). According to the reviewer’s suggestion, we further analyzed the observed variance in Fig. 5 and isolated the effect of different dataset types. We created two new boxplots (Fig. 5), one containing only correlation coefficients where both datasets contain natural images (Fig. R4F1) and one containing only coefficients for specialized+structured datasets (Fig. R4F2). The main observation remains the same: irrespective of the dataset selection, the training objective is a major driver for representational similarity consistency. However, we observe small differences such as a smaller minimum correlation for natural images  (see IN1k/XLarge & IN1k/Large) and a slightly larger variance for specialized and structure data (Fig. R4F2 vs Fig. R4F1). We will add the new plots to the camera-ready version’s appendix.
>
> Second, we will address the relevance of sigma for CKA values computed with RBF kernel. We normalize all feature representations, leading to comparable distances over datasets. However, we agree that the strong correlation between CKA RBF 0.4 and CKA linear suggests that $\sigma=0.4$ (partly) captures global similarity structures. Fig. 9 shows small differences in the upper right corner; however, we agree that the difference between CKA linear and CKA RBF 0.2 is more pronounced. We extend our correlation plot (Fig. 3) by CKA RBF 0.2 as shown in Fig. R4F3. As expected, CKA RBF 0.2 differs more significantly from CKA linear. Based on this evidence, we agree with your perspective that RBF 0.4 still captures some global similarity while RBF 0.2 is a better candidate for analyzing local similarity. Thus, we repeated our experiments with the local kernel (see Fig. R4F4 and R4F5). We observe the same overall pattern, indicating that the objective is relevant for similarity consistency, while network architecture is less important. This analysis reveals two interesting observations. For local similarity, the supervised models are more consistent than the Img-Txt models, which are better at recovering global structure. In addition, models trained on IN-21k are more consistent than models trained on IN-1k, of which the latter set is more consistent in their global structure. IN21k contains substantially more classes (21.843) and a higher percentage of the classes are leaf nodes in the WordNet tree (76.71% vs 65% for IN1k), representing more fine-grained entities. The representation must contain more fine-grained details to distinguish these classes, dominating local similarities. We will include these observations in Appx. F of the camera-ready version.
>
> Third, we investigate the stability of CKA values across different subsets. Fig. 10 shows that CKA linear is stable with 5k samples, showing smaller variance when increasing to 10k samples. To further analyze this stability, we performed bootstrapping over 500 subsets of IN-1k features, each containing 10k samples (as suggested), for 6 example models (the anchor models from Appx. G). Fig. R4F6 and the narrow confidence intervals (CI) in Tab. R4T1 confirm the stability of CKA linear values, demonstrating minimal variation across bootstrapped subsets. For RBF 0.2, Fig. 10 showed instability with a subset of size 10k, while 30k samples are more robust. We verified this with another bootstrapping experiment, Fig. R4F7, and the CIs in Tab. R4T2 showed a larger variance in CKA values. This suggests that local similarity depends more on the specific stimuli than global similarity. This is not surprising as stimulus-specific fine-grained details drive local similarity measurements. To mitigate this effect, we increased the number of samples for experiments analyzing local similarity from 10k to 30k when the dataset size permitted.
>
> Last, we agree that the t-SNE Figure helps us understand our categorization and will move this visualization from the appendix to the main part.
>
> [1] Zhai, Xiaohua, et al. "A large-scale study of representation learning with the visual task adaptation benchmark." arXiv preprint arXiv:1910.04867 (2019).

---

### Official Review · Reviewer_ZtBX · 2025-03-13

**Overall Recommendation:** 4

**Summary:**

The paper proposes a way of measuring the consistency of pairwise similarities across datasets and transferability of similarities between them. The authors provide many observations regarding these aspects.

## update after rebuttal

The authors provided some additional discussions and results, which further strengthened my belief that they deserve the high score I initially gave (accept). I hope that as promised the authors will incorporate the main points of their answers into their camera-ready version.

**Claims And Evidence:**

The claims made in the submission are supported by clear and convincing evidence.

**Essential References Not Discussed:**

The works [1] and [2] could be cited in the paper. They leverage CKA/RSA and task similarities for transfer learning.

[1] Borup, Kenneth, Cheng Perng Phoo, and Bharath Hariharan. "Distilling from similar tasks for transfer learning on a budget." Proceedings of the IEEE/CVF International Conference on Computer Vision. 2023.

[2] Dwivedi, Kshitij, and Gemma Roig. "Representation similarity analysis for efficient task taxonomy & transfer learning." Proceedings of the IEEE/CVF Conference on Computer Vision and Pattern Recognition. 2019.

**Experimental Designs Or Analyses:**

- The authors evaluated numerous models on many datasets. They also analyzed whether the results are not due to the similarity measure of choice. In my opinion, the experiments are convincing. (+)
- In Section 4.3 and Fig. 4, it is interesting that some SSL models lie very close to the text-image models. Could the authors check what SSL models cluster with text-image models and try to explain why? (question Q1)
- As the authors use models trained on general purpose datasets (such as ImageNet), it would be nice to present some more qualitative results for a dataset such as EuroSAT due to the fact that this dataset also presents a domain with a large domain gap to the general-purpose datasets (e.g. Fig. 2, 4). Similarly, it would be nice to include some examples for the DTD dataset, as it focuses on textures. (Q2)
- The authors should better discuss the inconsistencies between relative similarities in Section 4.6 (e.g. ImageNet and CIFAR10/100) - Q3.

**Methods And Evaluation Criteria:**

The methods are adequate for the problem (RSA, CKA). I like the simplicity and clear formulation of the framework. Also the datasets and models are adequate.

**Other Comments Or Suggestions:**

As mentioned before, the authors could once more review their figures and add to their descriptions some better explanations of the contents, especially for Fig. 5 (a brief description of what the labels on the images mean).

**Other Strengths And Weaknesses:**

Weaknesses:
- Some captions of the figures could be more informative. E.g. In Fig. 2, it would be nice to add the legend of the colors used for different boxes (it is only done in the text)
- In Fig. 5, some terms are difficult to understand (middle, bottom row) i it would be useful to briefly describe what the authors mean by Large etc. when it comes to the dataset size and similarly for model sizes.

Strengths:
- The authors compare different similarity metrics to minimize the possible impact of a given similarity measure on the results.
- It is a useful combination of the existing methods.

**Questions For Authors:**

**Q1**: Could the authors check what SSL models are placed close to the text-image models and try to explain why? (Fig. 4)

**Q2**: Would it be possible to add additional results to Fig. 2 and 4 (or somewhere in the Appendix) including the EuroSAT/DTD datasets (as an example of other specialized/structured datasets - as such comparisons are the most interesting due to a domain gap between the general purpose and specialized/structured datasets).

**Q3**: The authors state: “ Interestingly, ImageNet-1k exhibits a milder yet significant pattern of inconsistency. Within the multi-domain category, this is especially pronounced for CIFAR-10 and CIFAR-100” - do the authors think the reason is a significantly lower resolution of the images in the CIFAR datasets? Could the authors dig a little bit deeper to analyze why some datasets for visible clusters in Fig. 6?

**Relation To Broader Scientific Literature:**

Key contributions of the paper build on the previous methods (like CKA, RSA) and can be used as an extension of the existing testing procedures.

**Theoretical Claims:**

N/A - the nature of the paper is rather empirical.

---

> ### Author Rebuttal · Authors · 2025-03-31
>
> First, we thank the reviewer for their overall positive feedback and for pointing us to two papers that helped strengthen the integration of our work into the existing literature. We agree with their relevance to our work due to using Representational Similarity Analysis [RSA; 2] or other similarity measures [1] to (pre-)select (downstream) task-specific models. Therefore, we will cite them in the related work section of our manuscript’s camera-ready version: “[...] However, recent work demonstrated that representational similarities (e.g., measured via RSA) can be used to effectively select (downstream) task-specific models [1,2].”
>
>
> Second, we agree on the lack of clarity of some of the figure captions and will use the extra page in the camera-ready version to expand these captions to improve clarity, i.e., for Fig. 5.
>
>
> In addition, we thank the reviewer for posing three interesting questions, which we will elaborate on, providing additional results (R3F1-F5) in an anonymized [repository](https://anonymous.4open.science/r/rebuttal_similarity_consistency/README.md).
>
> **Q1**: _Which SSL models are close to Img-Text in Fig. 4, and why?_
>
>
> Fig. R3F1 zooms into the area of Img-Txt models for the three dataset combinations of Fig. 4 and labels individual model pairs. We observe that many SSL and image-text model pairs show high CKA values across the datasets, i.e., are located in the upper right corner and therefore in proximity. These models are similar within each objective. The SSL model pairs show high similarities as they are trained with similar datasets,  augmentations, and losses (e.g., BarlowTwins/VicReg and MoCov2/SimCLR). Some image-text models are quite similar as well. However, the proximity of these two model pair sets does not allow us to infer any direct relationship between them.  For this, we must consider the similarities of model pairs containing both model types, as shown in Fig. R3F2. The points, representing cross-type similarities between SSL and Img-Txt models (pink), have lower correlations than within-type pairs. This indicates that despite some individual SSL models appearing close to Img-Txt models, the overall relationship between these two categories of models is less strong and more variable.
>
> **Q2**: _Could Fig. 2 and 4 be provided for more specialized datasets?_
>
>
>
> We thank the reviewer for their suggestion and have decided to include additional figures extending Fig.2 with datasets of each category (Fig. R3F4) and Fig.4 containing the EuroSAT and DTD (Fig. R3F5).
>
> **Q3.1**: _Do you attribute the inconsistency patterns in CIFAR-10 and CIFAR-100 primarily to their significantly lower image resolution compared to IN-1k?_
>
>
> Yes, we indeed think that CIFAR-10/100’s small image size (32×32 pixels) restricts the representation space by severely limiting fine-grained details and surrounding contextual information. Appx. I shows clear differences in correlation coefficients between performance gaps (def. Sec. 4.7) and CKA values: low absolute correlation for IN-1k versus high for CIFAR. This suggests IN-1k’s high-resolution images (224×224 pixels) provide rich contextual cues that support diverse, well-performing representations, whereas CIFAR’s constrained resolution and minimal background details restrict the range of viable representations.
>
> **Q3.2**: _Can you elaborate on the underlying factors causing the visible clustering patterns observed for certain datasets in Fig. 6?_
>
>
> Looking more closely at the clustering patterns of Fig. 6, we observe:
> - CIFAR-10 and CIFAR-100 form a strong cluster, potentially due to their low-resolution format and similar categorical structures, showing also high consistency in the (Img-Txt, Sup).
> - The Breeds datasets, Caltech-101, Country-211, STL-10, and Pets cluster together based on their similar domain properties, resolution profiles, and centered object compositions, showing also high consistency in the (Img-Txt, Sup).
> - Medical imaging datasets do not cluster together, potentially due to their fundamentally different visual patterns and domain-specific features (eye vs. tissue scans).
> - RESISC45 shows stronger correlations with structured datasets than with other specialized datasets (i.e., (Img-Txt, Sup)). This cross-category relationship might stem from satellite imagery's inherent structural properties—regular grid layouts, transportation networks, and geometric patterns—which create feature distributions resembling those in structured datasets. However, it differs from EuroSAT potentially due to RESISC45's higher resolution imagery, greater geographic diversity, and more diverse class set.
>
> A further analysis that isolates natural images from structured and specialized datasets can be found in the first part of Reviewer 9H2r's answer.
>
> We will incorporate the main points of these answers into our camera-ready version.

---

### Official Review · Reviewer_7cXA · 2025-03-13

**Overall Recommendation:** 3

**Summary:**

This paper is more of an analytical paper that analyze the cross-domain representation similarity among models trained with different objectives. The analytical framework is pretty simple as it is a combination of kernalized CKA and a spearman correlation measure. Methodology description is concise. Experiments show the results of analysis, most of the observations are inconclusive due to the confounded factors, but the some insights are interesting. E.g. SSL's representation similarity consistency is higher than that of supervised learning.

**Claims And Evidence:**

There is no clear claim of this paper given it is more about analysis. While the paper claimed the "framework" of analysis as part of its contribution. It is a combination of pretty standard and well known analytical tools. If that is the key claim of this paper, then the paper's novelty is very low.

I think the biggest contribution of this paper is about the insights it provided on understanding the representation similarity among models in the context of domain-transfer. However, the analysis seems lack depth given multiple observations are less conclusive ( due to the confounded factors). Probably the experimental settings could be adjust to minimize the effect of confounded factors. I think this paper is more suitable for being a position paper rather than a standard ICML submission given its claim is about advocating a particular research direction rather than proposing a new algorithm.

**Essential References Not Discussed:**

N/A

**Experimental Designs Or Analyses:**

Analysis looks well done. I have learned something from this work. The only complain is about inconclusive observations here and there.

**Methods And Evaluation Criteria:**

Method is clearly written and easy to follow. It does make sense to leverage CKI and spearman correlation to quantify the representation similarities.
There is no meta-evaluation criteria to quantify the method proposed (given the method itself is a set of measure). However, the experimental setup can be further tuned to reduce confounded observations.

**Other Comments Or Suggestions:**

I think the figure reference in section 4.3 is wrong.

**Other Strengths And Weaknesses:**

Strength:
1. Well written article. Descriptions are concise and clean.
2. Extending representation similarity research to cross-dataset setting, which looks interesting.
3. Some observations from the analysis is interesting in terms of justifying the need of SSL in many applications.

Weakness:
1. The depth of analysis is still shallow. It could be more interesting if the authors can design the analysis with a more precise control on factors.

**Questions For Authors:**

N/A

**Relation To Broader Scientific Literature:**

This research is one direct extension of representation similarity measure among models (or could be even model vs neuron recordings).

**Theoretical Claims:**

N/A, there is no theoretical claims I can see unless "SSL's representation similarity consistency is higher than that of supervised learning" is the claim, which is empirical not theoretical.

---

> ### Author Rebuttal · Authors · 2025-03-31
>
> We would like to thank the reviewer for the valuable feedback and appreciate the assessment of our manuscript as being _well-written_ and _interesting_ work.
>
> First, we agree that the individual components of our analysis framework are well-established rather than novel by themselves. We see this as a strength rather than a weakness because it allowed us to build a novel meta-framework relying on the methodological soundness of the individual components without introducing subcomponents that still need to be validated by the research community. To the best of our knowledge, our work is the first to propose a structured way of using these components to characterize how similarity relationships vary across datasets. Our main focus lies on introducing and formalizing _similarity consistency_ and conducting a large-scale analysis of this measure across 64 different vision models and 23 datasets.
> We identify that the learning objective is a crucial driver of similarity consistency, and architectural specifications before training (architecture type and size) are less relevant.
>
> Second, while confounding factors cannot be avoided when evaluating the limited set of large, well-trained general vision models, we tried to unravel some confounders in Appx. G and H.
> - In Appx. G, we selected two anchor architectures (ResNet-50 and ViT-L) and systematically varied the training objective (Sup., SSL, Img-Txt) while keeping the architecture type and model size constant, resulting in six models. This allowed us to isolate the effect of the objective in a more limited but controlled setting. We observed that even here, the training objective appears to be driving similarity consistency.
> - In Appx. H, we investigate the role of training data on similarity consistency. To that end, we fixed the objective (supervised) and network architectures (AlexNet, DenseNet161, ResNet18, and ResNet50) and varied the training dataset (general-purpose vs. domain-specific). Here, we identified higher consistency in models trained on a domain-specific dataset (Places365), most likely due to a more constrained solution space in comparison to general-purpose models trained on large-scale datasets. These findings complement our main analysis, which deliberately focused on general-purpose vision models trained on datasets with large semantic diversity.
>
> Last, we fixed the Fig. reference in Section 4.3. We intended to refer to Fig. 11 in the Appx. D. We will correct this in the camera-ready version by using the extra page to move this figure back to the main text.

---

### Official Review · Reviewer_f2Fe · 2025-03-13

**Overall Recommendation:** 3

**Summary:**

The paper sets out to challenge the Platonic representation hypothesis by reexamining similarities between representation of models using multiple datasets. Their key finding is that training objective is a dominant factor driving representations, as opposed to model architecture and model size.

**Claims And Evidence:**

The claim on objective function needs more support, as currently only SSL versus supervised objectives were examined; there are many other important objective functions—such as robustness to noise or corruptions objectives, generative modeling objectives (e.g., diffusion models), masked image modeling, and reinforcement learning-based objectives—that were not included in the analysis. Without evaluating models trained on these diverse objectives, it is difficult to generalize the conclusion that the objective function is the primary driver of representational similarity consistency.

**Essential References Not Discussed:**

None spotted

**Experimental Designs Or Analyses:**

Experimental design is sound and standard for measuring similarity of representations.

**Methods And Evaluation Criteria:**

A key methodological limitation is that the study focuses exclusively on representations extracted from the final layers of the models (ref table in supplementary: the penultimate layer for supervised models, the average pooling layer for SSL models, and the image encoder output for image-text models). While this is a common practice, it risks biasing the analysis toward the influence of the objective function, since final-layer representations are often more task-specific and reflect the model’s training objective.

**Other Comments Or Suggestions:**

None

**Other Strengths And Weaknesses:**

The question and the approach are not novel so the work can really benefit from broadening the evaluations to more models to make a conclusive claim, or refine the claims to the actual results.

**Questions For Authors:**

If possible, could you also report the analysis results on a few internal layers, close to the middle of processing in each model? Do you expect to see more or less the same results?

**Relation To Broader Scientific Literature:**

The question of what drives similarity (or dissimilarity) of representations is important for many fields and it directly engages with many recent works on the topic.

**Theoretical Claims:**

No theoretical claims

---

> ### Author Rebuttal · Authors · 2025-03-31
>
> We would like to thank the reviewer for their helpful suggestions and for acknowledging the _relevance of our work_ and the _soundness of our experiments_. We believe that following the reviewer’s suggestions allowed us to notably improve our analyses. Two points stood out in particular, which we address in detail below. We added all additional figures (R1F1-F5) to an anonymized [repository](https://anonymous.4open.science/r/rebuttal_similarity_consistency/README.md)  and refer to the specific Figures detailed in the README.md.
>
> First, we would like to clarify the range of the covered training objectives. In representation learning for computer vision, self-supervision (SSL) and image-text alignment are currently the state-of-the-art approaches. Therefore, we selected these two categories alongside supervised learning. The SSL group contains a diverse set of objectives, including self-supervised contrastive losses (such as SimCLR), the mentioned _masked image modeling_ (MAE), self-distillation (DINO), extended self-distillation (DINOv2), but also pretext-task-based (Jigsaw, RotNet), redundancy-reduction (BarlowTwins) and clustering-based (SwAV) losses*. The total set of 64 models contains (most) SOTA models for representation learning in vision. For this diverse set, we identified the training objective as a crucial factor for the consistency of representational similarity, while model architecture and size appear less relevant.
>
> As we identified the training objective as a main driving factor for similarity consistency, we are convinced that due to its flexibility, our framework can, in future work, easily be applied to analyze other training objectives, e.g., as recommended, testing the effect of robustness losses on the similarity structure of out-of-distribution data.
>
> Second, we would like to address the effect of taking the _final layer_ of the model by evaluating the representational similarity consistency on intermediate layers. In our work, we followed the standard procedure of extracting features of the _final layer_ (or _penultimate layer_, for classification models), which is commonly referred to as the _representation_ [1]. We remark that these representations are of special interest because they are the ones used for downstream tasks. However, we agree with the reviewer on the potential role of layer choice for our findings, as mentioned in our discussion section.
> Our proposed analysis framework does _not depend on specific layers_; it can also be applied to intermediate layers. Therefore, we followed the reviewer's suggestion and repeated the consistency analysis for the middle layers of _a large subset of our transformer models_. We omitted CNN-based models, as middle-layer extraction is less clear when representations depend on spatial location. We remain consistent in our extraction method across layers for the transformer models: If the model's original representation was derived from the classifier token, we also extracted the classifier token from intermediate layers. Otherwise, we applied avg. pooling. Fig. R1F1 and R1F2 show similarity matrices, analogous to Fig. 2. Here, model representations tend to be more similar across models of the same type. W.r.t. the consistency of similarities, we observe a lower median and larger standard deviation of consistency of representational similarities across all model pairs (grey bar) in Fig. R1F5 compared to Fig. 5. While the variances are relatively large for the training data and model size categories, we see above-median consistency for within-objective model pairs.
> We speculate that higher consistencies within training objectives in intermediate layers indicate that the training objective has a stronger influence on representational structures already early in the network. For example, supervised models may form structurally similar lower-level representations but are less constrained in the organization of their representations close to the classification output.
>
> In conclusion, our analyses of intermediate layers support the finding that the training objective is a main influence on representational similarity consistency, though more experiments would be needed to fully characterize layer-specific effects. We thank the reviewer for their insightful comments that have strengthened our analysis.
>
> \* References to the models can be found in the main paper, in Tab. 2.
>
> [1] Kornblith, Simon, et al. "Do Better ImageNet Models Transfer Better?" CVPR, 2019.

---

### Decision · Program_Chairs · 2025-05-01

**Decision:**

Accept (poster)

**Comment:**

This manuscript investigates why models of different architectures, even ones trained on different modalities, are similar to one another. It concludes that architecture and model size have little impact on model similarity, while the training objective has a significant impact. This is a topic of wide interest to the community at the moment. The reviewers found the manuscript to be interesting, well-written, and the experiments well-executed. One might even imagine practical applications of this work, where one wants to develop maximally different networks to solve problems in different ways; the insights here would help.